# Branched ubiquitin chain binding and deubiquitination by UCH37 facilitate proteasome clearance of stress-induced inclusions

**Aixin Song[1], Zachary Hazlett[1], Dulith Abeykoon[2], Jeremy Dortch[1], Andrew Dillon[1], Justin Curtiss[1], Sarah Bollinger Martinez[1], Christopher P Hill[3], Clinton Yu[4], Lan Huang[4], David Fushman[2], Robert E Cohen[1], Tingting Yao[1]***

[1]Department of Biochemistry and Molecular Biology, Colorado State University, Fort Collins, United States; [2]Department of Chemistry and Biochemistry, University of Maryland, College Park, United States; [3]Department of Biochemistry, University of Utah School of Medicine, Salt Lake City, United States; [4]Department of Physiology and Biophysics, University of California, Irvine, Irvine, United States

**Abstract** UCH37, also known as UCHL5, is a highly conserved deubiquitinating enzyme (DUB) that associates with the 26S proteasome. Recently, it was reported that UCH37 activity is stimulated by branched ubiquitin (Ub) chain architectures. To understand how UCH37 achieves its unique debranching specificity, we performed biochemical and Nuclear Magnetic Resonance (NMR) structural analyses and found that UCH37 is activated by contacts with the hydrophobic patches of both distal Ubs that emanate from a branched Ub. In addition, RPN13, which recruits UCH37 to the proteasome, further enhances branched-chain specificity by restricting linear Ub chains from having access to the UCH37 active site. In cultured human cells under conditions of proteolytic stress, we show that substrate clearance by the proteasome is promoted by both binding and deubiquitination of branched polyubiquitin by UCH37. Proteasomes containing UCH37(C88A), which is catalytically inactive, aberrantly retain polyubiquitinated species as well as the RAD23B substrate shuttle factor, suggesting a defect in recycling of the proteasome for the next round of substrate processing. These findings provide a foundation to understand how proteasome degradation of substrates modified by a unique Ub chain architecture is aided by a DUB.

## Editor's evaluation

This study nicely addresses the role of a deubiquitylating enzyme (UCH37) in facilitating proteasomal clearance of branched polyubiquitylated substrates. Using a wide-range of chemical biological, biophysical and cell biological techniques, the authors convincingly demonstrate recognition of branched ubiquitin trimers and further demonstrate that mutations of the deubiquitylating enzyme lead to the formation of proteasomal foci in cells that are rich in polyubiquitinated species, presumably due to the loss of debranching activity. Overall, this excellent study adds to our understanding of UCH37 function, especially with regard to the newly observed phenomenon of reversible proteasome aggregation in cells.

*For correspondence:
tingting.yao@colostate.edu

## Introduction

In eukaryotes, the ubiquitin (Ub)–proteasome system is responsible for regulated protein degradation that maintains protein homeostasis and plays a major role in the defense against proteolytic stresses. Ub in the form of a variety of polyUb chains serves as a signal that targets unwanted proteins to the 26S proteasome where the proteins are unfolded and translocated into the interior chamber of the 20S proteasome where they are degraded. During this process, Ub is removed and recycled by deubiquitinating enzymes (DUBs). The timing of Ub removal is critical as premature deubiquitination can lead to unintended release of the substrate, whereas failure to remove Ub can inhibit substrate translocation. Additionally, polyUb chains released from the substrates need to be further disassembled in order to minimize competition with new substrates awaiting processing by the proteasome. Coordination of these actions relies on three DUBs associated with the proteasome: RPN11, USP14, and UCH37. Each of these DUBs has a nonredundant role in proteasome function. In recent years, significant progress has been made in our understanding of how RPN11 and USP14 are regulated to coordinate with substrate processing on the 26S proteasome, a role of UCH37 was discovered only recently (*Deol et al., 2020*).

Proteasome-associated DUBs can either promote or inhibit substrate degradation (*Lee et al., 2011*). RPN11 (also called POH1) is a stoichiometric subunit of the 19S regulatory particle (RP) of the proteasome; it is the only essential DUB in *Saccharomyces cerevisiae*. RPN11 activity is coupled to substrate degradation by removing polyUb chains *en bloc* as the substrate is translocated through a narrow gate into the proteolytic chamber of the 20S proteasome (*Yao and Cohen, 2002*; *Verma et al., 2002*; *Worden et al., 2017*). USP14 (UBP6 in yeast) is a dissociable subunit of the 19S RP (*Leggett et al., 2002*). It is activated upon forming a complex with the RP and removes Ub from substrates that are ubiquitinated at multiple sites (*Lee et al., 2016*). In vivo and in vitro evidence suggest that USP14/UBP6 inhibits proteasome degradation via both catalytic and noncatalytic mechanisms (*Lee et al., 2016*; *Lee et al., 2010*; *Hanna et al., 2006*; *Bashore et al., 2015*). In contrast to RPN11, USP14 can act before substrate is translocated into the 20S core particle and committed to degradation. Inhibition of proteasome-associated DUBs have broad effects on cellular proteostasis (*Li et al., 2017*; *D'Arcy et al., 2011*), highlighting the therapeutic potential of targeting proteasomal DUBs.

UCH37 (also known as UCHL5) is a highly conserved DUB found from fission yeast to humans, but absent in the budding yeast *S. cerevisiae*. It has a catalytic domain characteristic of the UCH family of DUBs and a unique C-terminal domain (CTD) that mediates interactions with two binding partners, RPN13 (*Hamazaki et al., 2006*; *Yao et al., 2006*; *Qiu et al., 2006*) and NFRKB. We previously reported that RPN13 (also known as ADRM1) recruits UCH37 to the 19S RP of the proteasome whereas NFRKB recruits it to the INO80 chromatin remodeler (*Yao et al., 2008*). Remarkably, whereas UCH37 is activated upon binding RPN13, it is held inactive in the complex with INO80. We and others have solved the crystal structures of UCH37 in both the activated and repressed states (*Vander Linden et al., 2015*; *Sahtoe et al., 2015*). These structures revealed that, whereas RPN13 stabilizes UCH37 in a conformation that promotes Ub binding, NFRKB represses UCH37 by occluding the Ub-binding site. Despite that these structures provided clear understanding of how UCH37 activity is controlled in different contexts, they did not reveal how UCH37 contributes to the functions of the proteasome or INO80 complex. Deletion of UCH37 in mice is embryonic lethal, although the underlying cause is unclear (*Al-Shami et al., 2010*). Cellular functions that have been associated with UCH37 include TGFβ signaling (*Wicks et al., 2005*), Hedgehog signaling (*Zhou et al., 2018*), DNA repair (*Nishi et al., 2014*), and cell cycle regulation (*Randles et al., 2016*); in these cases and others (*Li et al., 2019*; *Jacobson et al., 2014*) UCH37 was reported to either promote or inhibit degradation of a proposed substrate. However, whether the effects seen on specific substrates were direct or indirect were not resolved, and therefore UCH37's role in proteasome-mediated degradation remains enigmatic.

Among the variety of polyUb polymers in the cell, 10–20% have a branched architecture (*Swatek et al., 2019*) in which more than one Ub is attached to a single Ub acceptor. Branched polyUb chains have been reported to have a variety of functions (*Haakonsen and Rape, 2019*); most notably, polyUb branching enhances substrate degradation by the proteasome (*Yau et al., 2017*; *Meyer and Rape, 2014*). Recently, Deol et al. revealed that UCH37 has a K48-linkage-specific debranching activity that promotes proteasomal degradation of substrates modified with branched polyUb (*Deol et al., 2020*). Here, we set out to understand how the debranching specificity is achieved and how such an activity might contribute to proteasome function in vivo. Our work demonstrates that UCH37 binds to

branched polyUb by engaging the hydrophobic patches on both distal Ubs at a branch point. RPN13 further enhances the branched-chain specificity by restricting UCH37 from engaging linear polyUb chains. In cells, polyUb conjugates that accumulate upon proteolytic stresses are greatly enhanced by the loss of UCH37 activity. We show that both binding and deubiquitination of branched polyUb by UCH37 facilitate proteasome-dependent clearance of stress-induced inclusions.

## Results

### UCH37 and the UCH37–RPN13$^C$ complex prefer branched polyUb substrates

It was recently reported that UCH37 DUB activity with polyUb substrates is stimulated by branched-chain architectures (*Deol et al., 2020*). To better understand this unusual specificity, we generated homotypic and heterotypic Ub$_3$ chains containing all-native isopeptide linkages and having linear or branched architectures (for nomenclature, see *Figure 1—figure supplement 1A*). We also compared deubiquitination with UCH37 alone or in a complex with the minimal binding domain from RPN13 (RPN13$^C$, amino acid 285–407 *Vander Linden et al., 2015*); also known as the DEUBAD domain (*Sanchez-Pulido et al., 2012*; *Figure 1A, B*). By quantifying the release of the Ub$_2$ and Ub$_1$ products (*Figure 1C*), we drew the following conclusions: (1) The branched Ub$_3$ chains are strongly preferred as substrates (i.e., 10- to 100-fold faster hydrolysis) over their linear counterparts (e.g., [Ub]$_2$–$^{6,48}$Ub versus Ub–$^6$Ub–$^{48}$Ub or Ub–$^{48}$Ub–$^6$Ub). (2) Among the branched Ub$_3$ substrates, UCH37 strongly prefers K6/K48 over K11/K48 or K48/K63 branched chains. (3) Whereas RPN13$^C$ enhances UCH37 hydrolysis of most polyUb substrates, it strongly inhibits activity with linear K48 Ub$_3$ (*Figure 1C*). (4) Consistently, we found that Ub$_2$ and Ub$_1$ products were always produced in a 2:1 ratio quantified by mass (*Figure 1C*), which is equivalent to a 1:1 molar ratio. Using linkage-specific anti-Ub antibodies, we found that only the K48 linkage in K6/K48-branched Ub$_3$ was cleaved (*Figure 1—figure supplement 1B*), indicative of a 'debranching' activity that thus far is unique to UCH37. This observation is consistent with a previous report that UCH37 exclusively cleaves the K48 linkages in branched polyUb (*Deol et al., 2020*).

By promoting hydrolysis of branched polyUb chains while suppressing disassembly of linear K48 Ub chains, RPN13$^C$ potentiates the debranching specificity of UCH37. In the crystal structures of UCH37 in complex with Ub and RPN13$^C$, we and others have observed that RPN13$^C$ interacts with two distinct regions of UCH37: in addition to binding the UCH37$^{CTD}$, RPN13$^C$ also makes contacts with UCH37 active site crossover loop (ASCL) residues M148 and F149. These latter contacts restrict the conformation of the ASCL and promote UbAMC hydrolysis (*Vander Linden et al., 2015*; *Sahtoe et al., 2015*; *Figure 1—figure supplement 1C*). Mutations of ASCL residues M148 and F149 to alanines (AA) or aspartates (DD) had little effect on hydrolysis of the branched [Ub]$_2$–$^{6,48}$Ub substrate, but they abolished the ability of RPN13$^C$ to suppress linear K48 Ub$_3$ hydrolysis (*Figure 1D, E*, *Figure 1—figure supplement 1D*). Thus, RPN13$^C$ contacts with the UCH37 ASCL play an unexpected role to enhance UCH37 substrate specificity for branched polyUb.

### UCH37–RPN13$^C$ preferentially binds and deubiquitinates K6/K48-branched Ub$_3$

To better understand the factors contributing to UCH37 substrate specificity, we measured the binding affinities of catalytically inactive UCH37(C88S)–RPN13$^C$ with linear and branched Ub chains using microscale thermophoresis (*Figure 2A* and *Figure 2—figure supplement 1A*). UCH37(C88S)–RPN13$^C$ showed highest affinity for [Ub]$_2$–$^{6,48}$Ub ($K_D$ = 4.9 μM) whereas its affinities for [Ub]$_2$–$^{11,48}$Ub and [Ub]$_2$–$^{48,63}$Ub were substantially weaker and similar to those for linear Ub$_3$. Thus, substrate binding only partially accounts for UCH37 specificity.

In order to determine the kinetics of debranching, we took advantage of a previously developed free Ub sensor, tUI, which specifically and tightly ($K_D$ = 10$^{-10}$ M) binds to mono or polyUb that has a free C-terminus (G76) (*Choi et al., 2019*). Upon cleavage of the K48 linkage in a branched Ub$_3$, free Ub is released and captured quantitatively by Atto532-labeled tUI, leading to a fluorescence increase that can be monitored in real time (*Figure 2B*). To prevent binding of the substrate by Atto532-tUI, the proximal Ub in each substrate was modified by deletion of the C-terminal diglycine (UbΔGG). Using this free Ub sensor-based assay, we determined the $K_M$ and $k_{cat}$ for debranching of K6/K48, K11/K48, and K48/K63 Ub$_3$ by UCH37–RPN13$^C$ (*Figure 2C*). Comparison of the $K_M$ values shows that

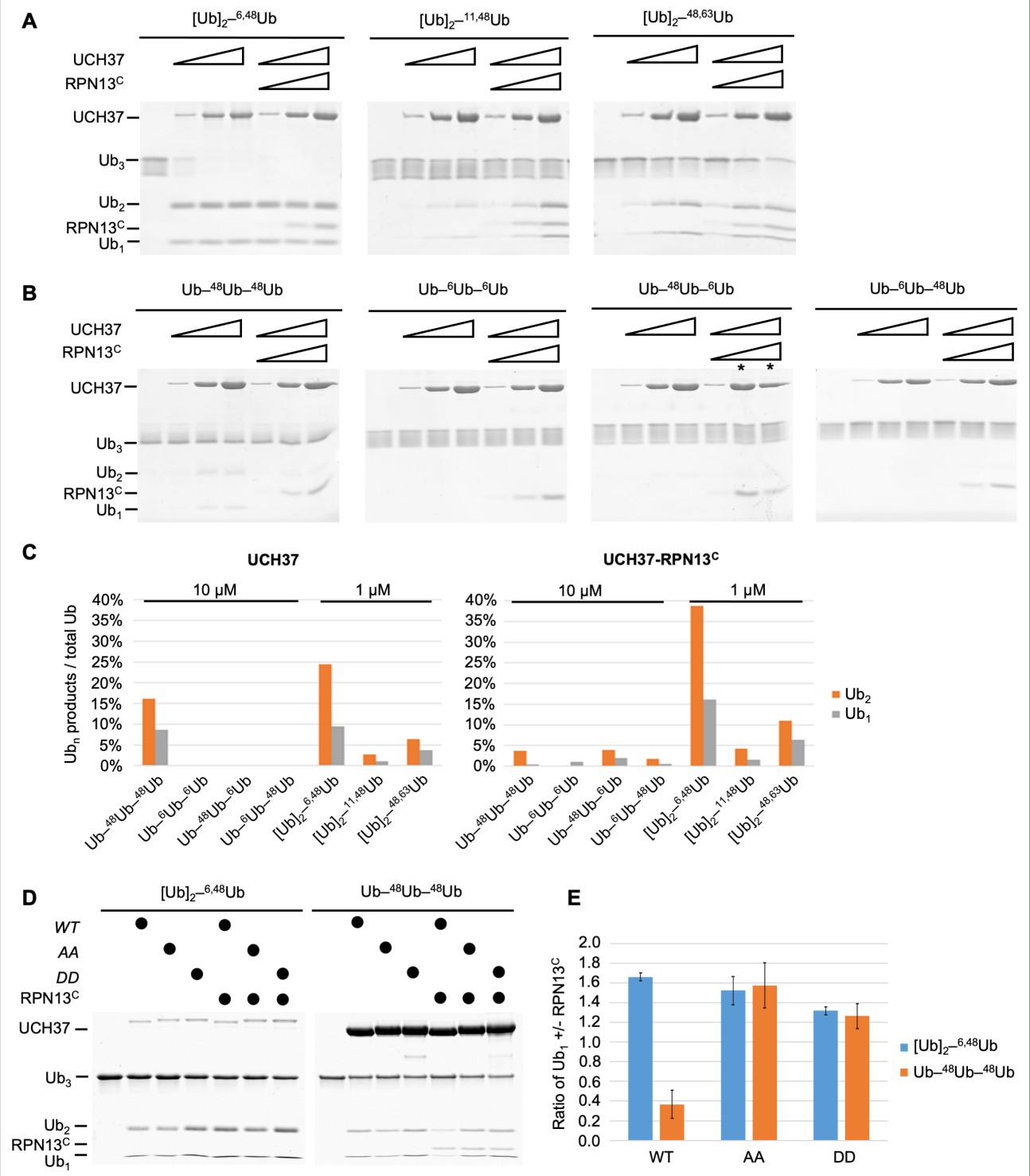

**Figure 1.** UCH37–RPN13$^C$ preferentially cleaves branched polyUb chains. Branched (**A**) or linear (**B**) Ub$_3$ substrates (5 μM) with the indicated Ub–Ub linkages were incubated for 1 hr at 37°C with 1, 5, or 10 μM His-TEV-UCH37, with or without the addition of equimolar RPN13$^C$. Reactions were analyzed by sodium dodecyl sulphate-polyacrylamide gel electrophoresis (SDS–PAGE) and Coomassie staining. (**C**) Quantification of the Ub$_2$ and Ub$_1$ products from (**A**) and (**B**). For linear Ub$_3$ substrates, results from incubations with 10 μM enzyme are shown; for branched Ub$_3$ substrates, results from 1 μM enzyme are shown. (**D**) Wild-type (*WT*), M148A F149A (*AA*), or M148D F149D (*DD*) UCH37, alone or with the addition of equimolar of RPN13$^C$, were incubated with Ub$_3$ substrates for 2 hr at 37°C. *Left*, reactions contained 0.5 μM enzymes and 10 μM branched substrates. *Right*, reactions contained 10 μM enzymes and 5 μM linear Ub$_3$. (**E**) Quantification from (**D**), plotted as the ratio of Ub$_1$ produced in the presence over absence of RPN13$^C$. Mean ± standard deviation (SD) from two independent replicates are shown.

*Figure 1 continued on next page*

*Figure 1 continued*

The online version of this article includes the following source data and figure supplement(s) for figure 1:

**Source data 1.** Uncropped gels in *Figure 1*.

**Figure supplement 1.** UCH37–RPN13$^C$ preferentially cleaves branched polyUb chains.

**Figure supplement 1—source data 1.** Raw western data in supplement 1B.

UCH37–RPN13$^C$ binds K6/K48-branched Ub$_3$ in preference to K11/K48 or K48/K63 branched Ub$_3$. Unexpectedly, $k_{cat}$ was highest for K11/K48 Ub$_3$; this resulted in similar catalytic efficiencies ($k_{cat}/K_M$) of debranching by UCH37–RPN13$^C$ for K6/K48 and K11/K48 Ub$_3$, whereas K48/K63 Ub$_3$ cleavage efficiency is 12–16 times lower.

We noticed that the $k_{cat}$ we measured for K6/K48 Ub$_3$ debranching at 30 °C is slower than what was reported by others, which varied from 3- to 12-fold faster at 37 °C, whereas the $K_M$ values are similar (*Deol et al., 2020*). To test if this discrepancy was due to artifacts from affinity tags or different methods of purification, we evaluated four different enzyme preparations by Ub-AMC hydrolysis (*Figure 2— figure supplement 1B*) and Ub$_3$ debranching (*Figure 2—figure supplement 1C, D*) assays. In neither assay were the results affected significantly by an N-terminal His-TEV tag on UCH37, whether RPN13$^C$ was added in trans or copurified with UCH37, or use of full-length RPN13 (RPN13$^{FL}$) versus RPN13$^C$.

## UCH37–RPN13$^C$ contacts the hydrophobic patches on both distal Ub units in a K6/K48-branched chain substrate

The K48-specific debranching by UCH37–RPN13$^C$ suggests that both distal Ub units in a branched Ub$_3$ substrate make unique contacts with the enzyme. To probe for those interactions, we incorporated $^{15}$N-labeled Ub as either the K6- or K48-linked distal Ub or the proximal Ub and performed NMR studies in the presence of the catalytically inactive UCH37(C88A)–RPN13$^C$ complex.

For the K48-linked distal Ub, we observed that the enzyme contacts the canonical Ub hydrophobic patch residues L8, I44, and V70 (*Figure 3A* and *Figure 3—figure supplement 1A*), consistent with the reported crystal structures of the UCH37–RPN13$^C$–Ub complex (*Vander Linden et al., 2015*; *Sahtoe et al., 2015*). Additionally, there were notable chemical shift perturbations (CSPs) and signal attenuations in the Ub α-helix (residues 23, 27, 29, 30, and 32–34) that did not correspond to any monoUb contacts in the UCH37–RPN13$^C$–Ub crystal structures. For the K6-linked distal Ub, we also observed strong signal attenuations of L8 and I44 (*Figure 3B* and *Figure 3—figure supplement 1B*). The interpretation of these NMR signal perturbations is complicated (*Castañeda et al., 2016*) by the well-known observation that the hydrophobic patch of a proximal Ub in a chain potentially interacts with hydrophobic patch residues of a neighboring distal Ub unit (*Cook et al., 1992*; *Varadan et al., 2002*; *Virdee et al., 2010*). In fact, NMR spectra revealed some effects on the hydrophobic residues in the proximal Ub (*Figure 3C* and *Figure 3—figure supplement 1C*), albeit the signal perturbations were not as severe as in the two distal Ubs. Thus, we sought to simplify the system by eliminating the hydrophobic patch on the proximal Ub. Upon addition of UCH37–RPN13$^C$, we did not observe significant CSPs with the mutated proximal Ub(L8A,I44A). The NMR spectra are similar to those obtained with UCH37–RPN13$^C$ added to monoUb (*Figure 3D* and *Figure 3—figure supplement 1D, E*). This could be due to either loss of intramolecular interactions between the Ub units in Ub$_3$, or loss of intermolecular interactions with UCH37–RPN13$^C$. Importantly, we found that a K6/K48 Ub$_3$ substrate with L8A,I44A mutations in the proximal Ub is debranched by UCH37–RPN13$^C$ at least as efficiently as the wild-type (WT) substrate (*Figure 3E*), indicating that the hydrophobic patch in the proximal Ub is not required for either binding or catalysis.

## The hydrophobic patch on the K6-linked distal Ub is required for debranching

To test whether the hydrophobic patch of K6-linked distal Ub is required for the debranching activity, we sought to replace it with a less hydrophobic or charged residue. However, poor utilization of Ub(L8A,I44A) for conjugation by the E1/UbcH5/NleL enzymes precluded direct substitution by this Ub mutant. As an alternative approach, we adopted the strategy used by *Cooper et al., 2009*. We synthesized K48-linked Ub$_2$ and then attached Ub(L8C) at K6 using E1/UbcH5/NleL; Ub(L8C) can be incorporated into chains by the ubiquitination enzymes and then modified with iodoacetic acid to

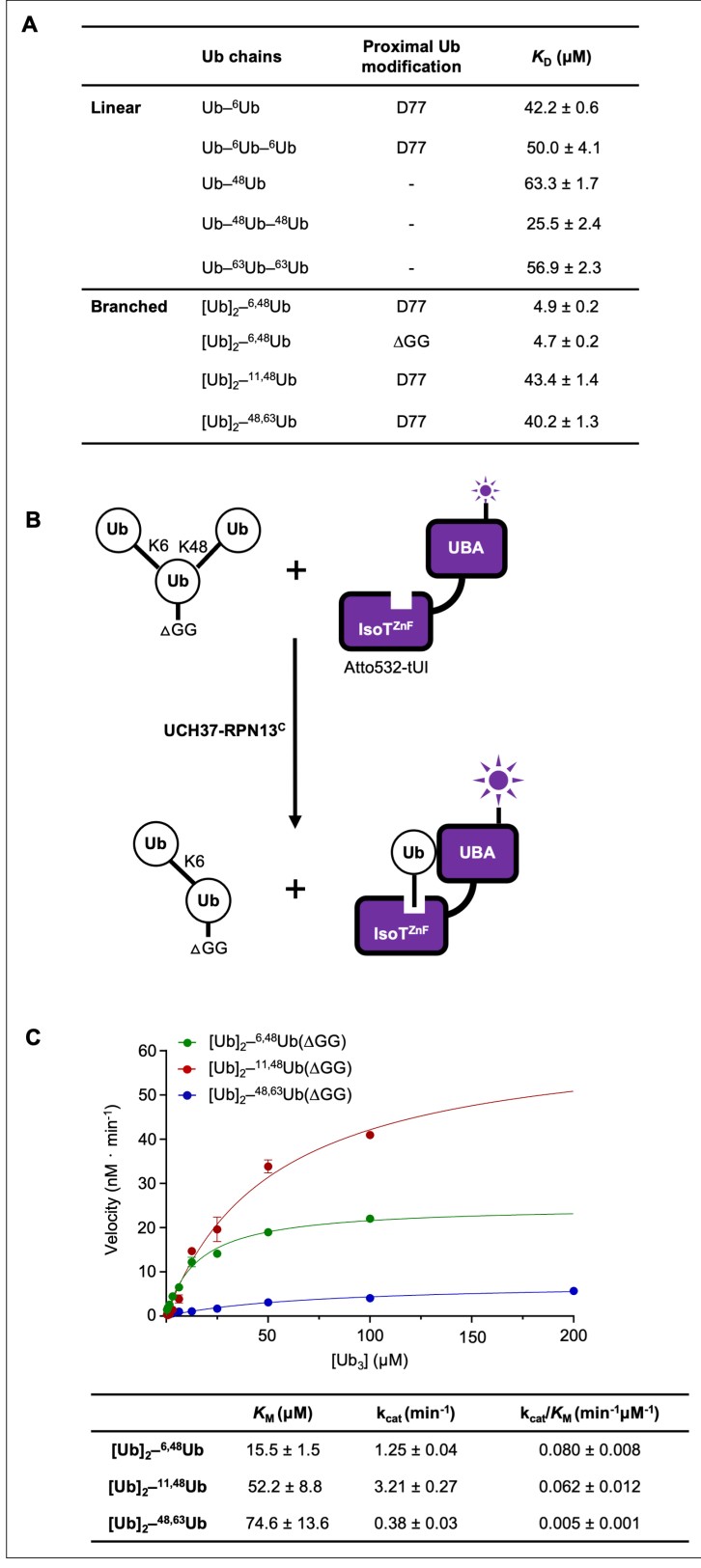

**Figure 2.** UCH37–RPN13[C] preferentially binds and deubiquitinates K6/K48-branched Ub₃. (**A**) Binding affinities between His-TEV-UCH37(C88S)–RPN13[C] and various polyUb chains were measured by microscale thermophoresis. Binding isotherms ( *Figure 2—figure supplement 1*) were fit with a single-site-binding model; best-fit $K_D$ values are shown with standard errors. (**B**) Schematic of the free Ub sensor-based deubiquitination assay. (**C**) Michaelis–

*Figure 2 continued on next page*

*Figure 2 continued*

Menten kinetics of branched $Ub_3$ hydrolysis by NS–UCH37–$RPN13^C$. The table shows best-fit $K_M$, $k_{cat}$, and $k_{cat}/K_M$ values with standard errors from two independent replicates.

The online version of this article includes the following source data and figure supplement(s) for figure 2:

**Figure supplement 1.** UCH37–$RPN13^C$ preferentially binds and deubiquitinates K6/K48-branched $Ub_3$.

**Figure supplement 1—source data 1.** Uncropped gels in supplement 1C.

---

create *S*-carboxymethylcysteine (L8Cmc), thereby adding a negative charge to the side chain. Using these variants of K6/K48 $Ub_3$, we found that L8C on the K6-linked distal Ub significantly inhibited debranching by UCH37–$RPN13^C$; moreover, debranching was inhibited even further by the L8Cmc derivative (*Figure 4A*).

In another approach to ablate the hydrophobic patch interaction, we utilized two variations of a dichloroacetone-based crosslinking strategy. First, Ub(G76C) was crosslinked to Ub–$^{48}$Ub(K6C) to produce Mimic1 (*Figure 4B*). The resultant nonhydrolyzable Ub–Ub crosslink is of similar length to a native isopeptide bond but contains an additional carboxylate (*Yin et al., 2000*). To eliminate the carboxylate, in a second variation we used intein chemistry to produce a $Ub_{75}$–cysteamine derivative, which was then crosslinked to Ub–$^{48}$Ub(K6C) to produce Mimic2 (*Figure 4B*). Interestingly, while with Mimic1 we observed a twofold reduced debranching rate by UCH37–$RPN13^C$, Mimic2 was debranched as efficiently as WT K6/K48 $Ub_3$ (*Figure 4C*). We then proceeded to install either WT or Ub(L8A,I44A) at K6 of the proximal Ub using the Mimic2 strategy. Whereas both native $Ub_3$ and WT Mimic2 $Ub_3$ were debranched efficiently, the L8A,I44A-containing Mimic2 was completely refractory to UCH37–$RPN13^C$ (*Figure 4D*). As a control, we showed that all the substrates were efficiently hydrolyzed by OTUB1, a K48-specific DUB.

Taken together, the results of these biochemical experiments demonstrate that UCH37–$RPN13^C$ requires hydrophobic patch residues of both distal Ub units of the branched $Ub_3$ substrate for its debranching activity. These results further support our interpretation of the NMR data as showing that, when bound to UCH37–$RPN13^C$, both distal Ub units use their hydrophobic patches to contact the enzyme.

## UCH37 activity regulates proteasome condensates upon proteolytic stress

In order to explore the physiological functions of UCH37, we generated a UCH37-knockout (KO) cell line by CRISPR from parental (P) HCT116 cells. The KO cell line was then complemented with three versions of UCH37: WT, catalytically inactive (C88A), and Ub-binding deficient (E34K, W36D, and I216D; referred to as EWI) (*Figure 5—figure supplement 1A*). Successful KO and reintroduction of UCH37 were confirmed by western blotting (*Figure 5—figure supplement 1B*). The EWI mutant was designed based on the crystal structure of the UCH37–$RPN13^C$–Ub complex (*Vander Linden et al., 2015*; *Sahtoe et al., 2015*; *Figure 1—figure supplement 1C*) and its reduced ability to bind Ub was confirmed (*Figure 6—figure supplement 1A*).

Osmotic stress was recently shown to trigger the formation of proteasome liquid–liquid phase-separated condensates that are thought to be centers of active degradation (*Yasuda et al., 2020*). As expected, upon osmotic stress, UCH37 localized to proteasome foci where K48-linked polyUb was also prominent (*Figure 5A*). We found that the KO cells accumulated significantly more proteasome foci in comparison with the parental cells (*Figure 5B*). We also observed increased accumulation of K11/K48-branched polyUb in the proteasome foci in KO cells (*Figure 5—figure supplement 1C*). In comparison with the parental cells, overexpression of WT UCH37 suppressed proteasome foci accumulation, whereas overexpression of C88A or EWI mutant UCH37 had the opposite effect (*Figure 5B*). Interestingly, although most proteasome foci appeared in the nuclei of the parental cells, there was also a marked increase in cytosolic foci in KO cells (*Figure 5C*). Similar accumulations of proteasome foci in KO cells were observed following oxidative stress (*Figure 5—figure supplement 1D*). These results strongly suggest that UCH37 DUB activity counteracts proteasome foci formation promoted by proteolytic stress. We noticed also that proteasome foci were readily detectable in KO cells even without sucrose treatment (*Figure 5D*), suggesting that the KO cells accumulate

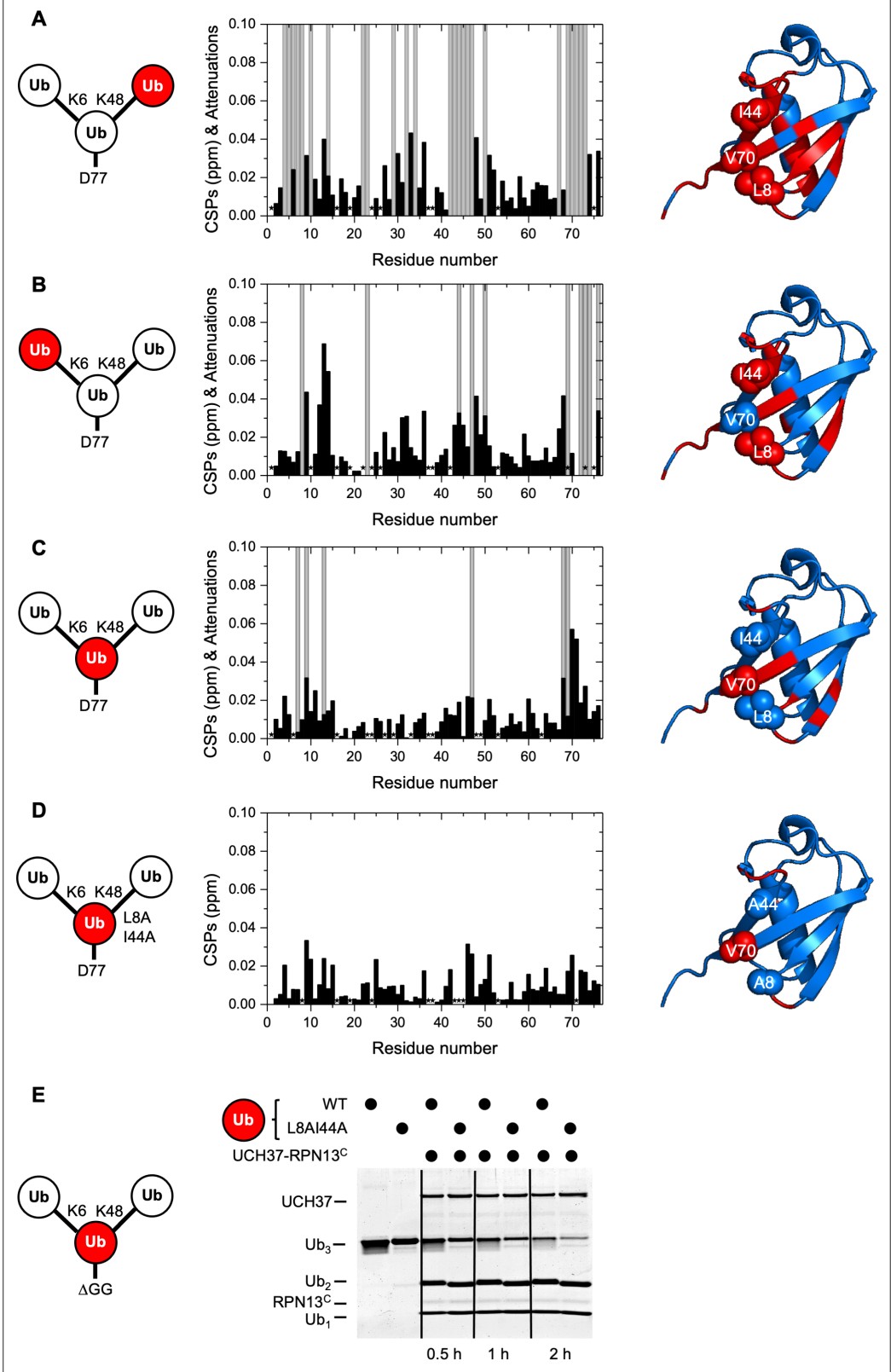

**Figure 3.** UCH37–RPN13[C] contacts the hydrophobic patches on both distal ubiquitin (Ub) units in a branched $Ub_3$ chain. Residue-specific perturbations of backbone amide NMR signals in the (**A**) K48-linked distal Ub, (**B**) K6-linked distal Ub, (**C**) the proximal Ub, and (**D**) mutated proximal Ub(L8A,I44A) in branched K6/K48-linked $Ub_3$ caused by the addition of 1.2 molar equivalents of copurified UCH37(C88A)–RPN13[C]. The NMR spectra are shown in

*Figure 3 continued on next page*

*Figure 3 continued*

**Figure 3—figure supplement 1**. Black bars represent chemical shift perturbations (CSPs, in ppm), gray bars mark residues exhibiting strong signal attenuations. Residues that were not observed or could not be unambiguously assigned/quantified due to signal overlap are marked with asterisks. Residues with strong signal attenuations or CSP >0.025 ppm are mapped (red) on the 3D structure of Ub (PDB code: 1UBQ); the hydrophobic patch residues are shown in sphere representation. (**E**) 1 µM UCH37–RPN13$^C$ was incubated with 5 µM substrate as indicated at 37 °C. Reaction products were analyzed by sodium dodecyl sulphate-polyacrylamide gel electrophoresis (SDS–PAGE) and Coomassie staining.

The online version of this article includes the following figure supplement(s) for figure 3:

**Source data 1.** Uncropped gel in *Figure 3E*.

**Figure supplement 1.** NMR spectra show unit-specific Ub$_3$ binding to UCH37–RPN13$^C$.

proteasome substrates under unperturbed conditions, which is consistent with a recent report (*Osei-Amponsa et al., 2020*). Indeed, analysis of the whole-cell lysates from the KO cells showed accumulation of K48 and K11/K48 polyUb both with and without osmotic stress (*Figure 5E, F*). This phenotype was partially rescued by overexpression of WT UCH37. These results suggest that the loss of UCH37 negatively impacts proteasome degradation capacity in cells.

## Ub, Ub–protein conjugates, and the RAD23B substrate shuttle receptor accumulate on proteasomes containing UCH37(C88A)

Under nonperturbed conditions, UCH37 dynamically interacts with the 19S RP (*Wang and Huang, 2008*; *Wang et al., 2007*) and, at steady state, appears in only a minor fraction of isolated mammalian proteasome complexes. To better understand how UCH37 activity affects proteasomal degradation, we focused on UCH37-containing proteasomes isolated from WT, C88A, and EWI cells. From the whole-cell lysates, it was apparent that C88A and EWI cells accumulate K48, K11/K48 and, to a lesser extent, total polyUb conjugates; this was the case both with and without osmotic stress (*Figure 6A*). These results are consistent with the phenotypes observed by proteasome foci counting (*Figure 5B*). UCH37-containing proteasomes were immunoprecipitated from these cells under low salt conditions to preserve weakly associated proteins (*Figure 6B*). Whereas the C88A proteasomes accumulated polyUb conjugates, surprisingly, the EWI proteasomes behaved similar to WT. To confirm these observations, we also created HEK293 cell lines that inducibly express different forms of UCH37 (*Figure 6—figure supplement 1A*). Again, polyUb accumulation was only observed with proteasomes containing C88A (lanes 7, 8, and 12). Enhanced polyUb association occurs in the context of the proteasome, not free UCH37, as UCH37ΔCTD, which cannot associate with RPN13, did not coimmunoprecipitate with either proteasomes or polyUb conjugates; this was observed with either WT or C88A UCH37ΔCTD (*Figure 6—figure supplement 1A*). UCH37 with the EWI and C88A mutations combined had only slightly increased amounts of associated polyUb conjugates in comparison with EWI alone, confirming that the EWI mutations effectively eliminated most, if not all, Ub binding to UCH37 (*Figure 6—figure supplement 1A*). We noticed that a fraction of C88A UCH37 were oligoubiquitinated at steady state. To address whether the ubiquitination status of UCH37 contributes to the phenotype observed with C88A proteasomes, we mutated the four primary ubiquitination sites (*Hornbeck et al., 2015*), K154, K158, K206, and K210, to arginines. The 4KR mutations substantially reduced the oligoubiquitinated species of C88A UCH37 but did not affect polyUb accumulation on C88A proteasomes (*Figure 6—figure supplement 1B*).

To quantify the amounts and types of (poly)Ub species associated with the UCH37-containing proteasomes, we used Atto532-labeled tUI in a previously developed protocol (*Choi et al., 2019*) to inventory free (i.e., unconjugated Ub or unanchored polyUb), activated (i.e., thioester form), and conjugated Ub species (*Figure 6C*). We found that both free and conjugated Ub species accumulated on C88A proteasomes at levels 4 times that of WT or EWI proteasomes. Additionally, the increased polyUb on C88A proteasomes was accompanied by increased association of the substrate shuttle receptor, RAD23B (*Figure 6D*). These results suggest that, although neither the C88A nor EWI form of UCH37 can catalyze deubiquitination, only the C88A proteasomes fail to clear Ub species efficiently.

Taking advantage of its GFP-tag, we examined the dynamics of UCH37 association with proteasomes directly in live cells. After sucrose treatment, we performed fluorescence recovery after photobleaching (FRAP) analyses of GFP-UCH37 in sucrose-induced proteasome foci. Strikingly, the average

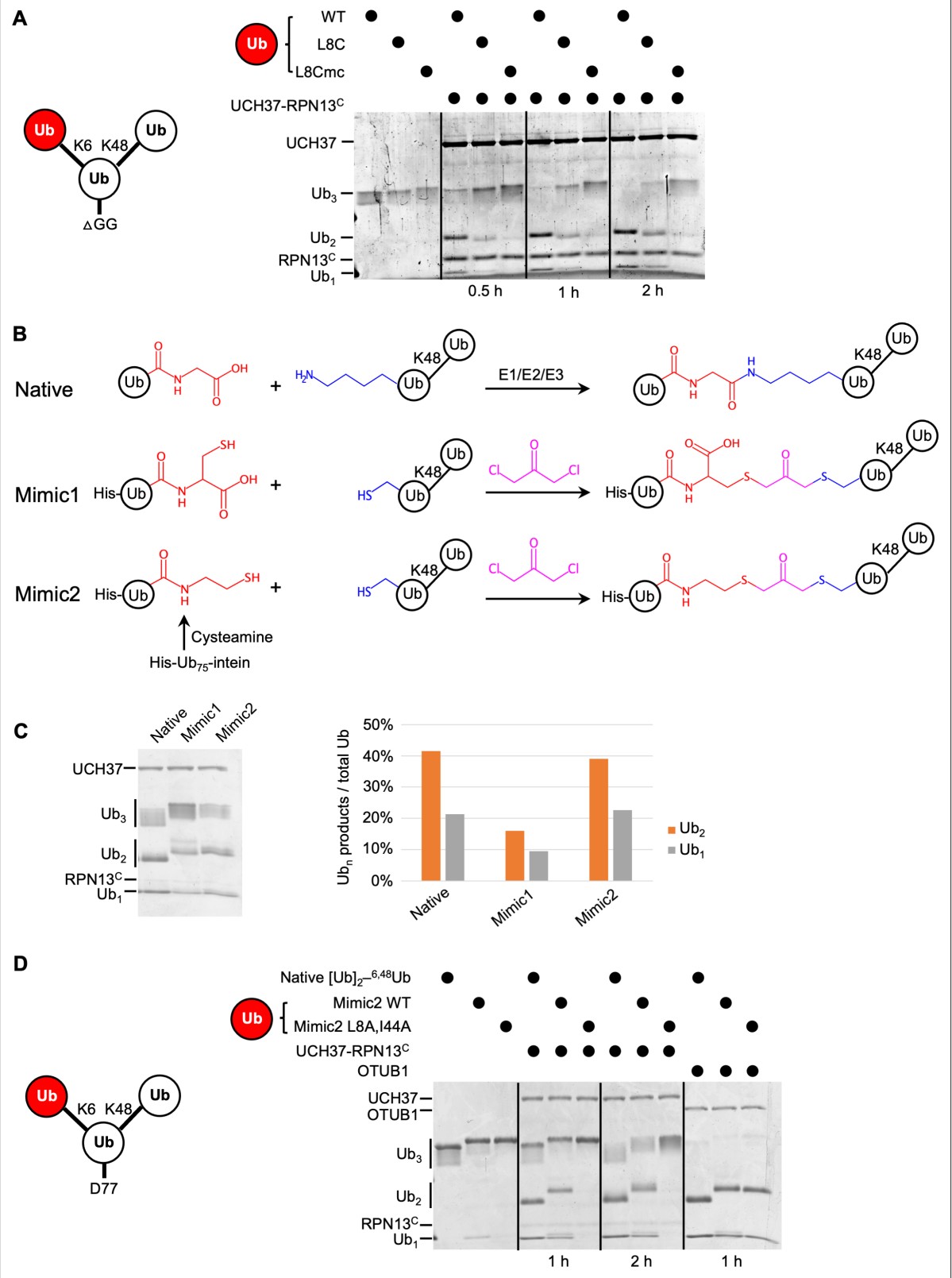

**Figure 4.** The hydrophobic patch on the K6-linked distal ubiquitin (Ub) is required for debranching. (**A**) 1 µM enzyme was incubated with 5 µM of the indicated substrate at 37 °C; at the times indicated, aliquots were taken and analyzed by sodium dodecyl sulphate-polyacrylamide gel electrophoresis (SDS–PAGE) and Coomassie staining. (**B**) Schematic showing assembly strategies and structures of branched K6/K48–$Ub_3$ and crosslinked mimics. (**C**) Comparison of native K6/K48–$Ub_3$ with mimics 1 and 2 in gel-based DUB assays using NS–UCH37–$RPN13^C$ as described in (**A**). Quantification of $Ub_2$ and

*Figure 4 continued on next page*

*Figure 4 continued*

$Ub_1$ products from the gel are plotted as shown. (**D**) Native K6/K48–$Ub_3$ or Mimic2 $Ub_3$ containing either wild-type or L8A,I44A mutant Ub crosslinked to Ub–$^{48}$Ub(K6C) were analyzed by gel-based DUB assays as described in (**A**).

The online version of this article includes the following figure supplement(s) for figure 4:

**Source data 1.** Uncropped gels in *Figure 4*.

FRAP recovery half-times for UCH37 WT (10.7 s) and EWI (7.2 s) were three to four times shorter than that of C88A (27.4 s) (*Figure 6E*). Given that the reported $t_{1/2}$ of proteasomes in sucrose-induced foci is 13.71 s (*Yasuda et al., 2020*), similar to that of UCH37 WT, these data suggest that C88A proteasomes in the condensates are significantly more static.

To determine the compositions of UCH37-containing proteasomes in an unbiased fashion, we performed tandem mass tag (TMT) mass spectrometry analyses of WT, C88S, and EWI proteasomes. For the majority of the constitutive proteasome RP and CP subunits, we did not find any differences (*Figure 6—figure supplement 1F*). However, among dynamically associated proteasome-interacting proteins (PIPs), we found that Ub (UBA52), RAD23B, PA28 (PSME1 and PSME2), USP14, and 19S assembly chaperones (PSMD5 and PSMD10) were noticeably enriched in the C88S and, to a lesser extent, EWI proteasomes (*Figure 6F*). The extent of Ub accumulation detected by TMT analysis is less than that detected by tUI-based assays (*Figure 6C*), most likely due to ratio suppression commonly observed in TMT-based quantification (*Ting et al., 2011*). To rule out that this is due to differences between UCH37 C88A and C88S mutants, we compared them in both pulldown assays and in vitro binding assays, and found that both mutants behaved similarly (*Figure 6—figure supplement 1C,D*).

The accumulation of RAD23B on C88A (or C88S) proteasomes raised the possibility that, among substrate shuttle proteins, RAD23B may uniquely prefer branched Ub chains; this prompted us to determine the binding affinities between full-length RAD23B and various forms of $Ub_3$. We found that RAD23B does not differentiate linear versus branched $Ub_3$ provided that a K48 linkage is present within the chain (*Figure 6—figure supplement 1E*).

## Discussion

UCH37 is the only DUB known to date that has a strong preference for branched polyUb chains. Our results have revealed multiple levels of control that contribute to the unique debranching specificity of UCH37. Like other UCH family of enzymes, the ASCL of UCH37 controls access to its active site. In the ground state, RPN13's interaction with the ASCL promotes a conformation that prevents linear polyUb chains from accessing the active site. By microscale thermophoresis, we determined that UCH37–RPN13$^C$ bound to K48- and K6-linked $Ub_2$ with similarly weak affinities. Additionally, mixed-linkage linear polyUb chains, such as Ub–$^6$Ub–$^{48}$Ub and Ub–$^{48}$Ub–$^6$Ub, are poor deubiquitination substrates. This likely reflects that the ASCL is positioned to block linear polyUb binding regardless of linkage. Thus, the ASCL functions as a gate and RPN13 as a gatekeeper to prevent unintended hydrolysis of linear polyUb chains. Upon encountering branched chains, UCH37 engages the hydrophobic patches of both distal Ub units; this would enhance affinity and, crucially, affect the displacement of the ASCL to allow access to the active site. The geometry of a branched polyUb chain likely plays a key role in substrate binding, position of the ASCL, and K48 specificity in debranching by UCH37–RPN13. Our data suggest that the K6/K48-branched $Ub_3$ presents a favorable configuration to engage and activate the enzyme. Given that we did not observe significant NMR CSPs of Ub residues around the K48 linkage on the proximal Ub, we favor a model in which the configuration of the two distal Ubs at the branch point directs specificity for cleavage of the K48 linkage. Other than K48, there are six lysine residues and one N-terminal amine where another distal Ub can be attached, thereby giving rise to a variety of possible branched $Ub_3$ structures. Our model predicts that only some of these possible structures are ideal substrates for UCH37–RPN13; thus, polyUb substrates with Ub units branching from different lysine residues will not be hydrolyzed with equal efficiencies. The finding that the efficiency of substrate disassembly is in the order of K6/K48 > K11/K48 >> K63/K48 supports this idea. Currently, we do not know if UCH37–RPN13 is active with branched polyUb lacking a K48 linkage.

The minimal branched polyUb chain has three Ub units. Using K6/K48 $Ub_3$ as a model, our NMR experiments suggest that UCH37 has at least two Ub-binding sites, each engaging the hydrophobic patch of a distal Ub unit in a branched chain. One of these sites, presumably the S1 site, is occupied

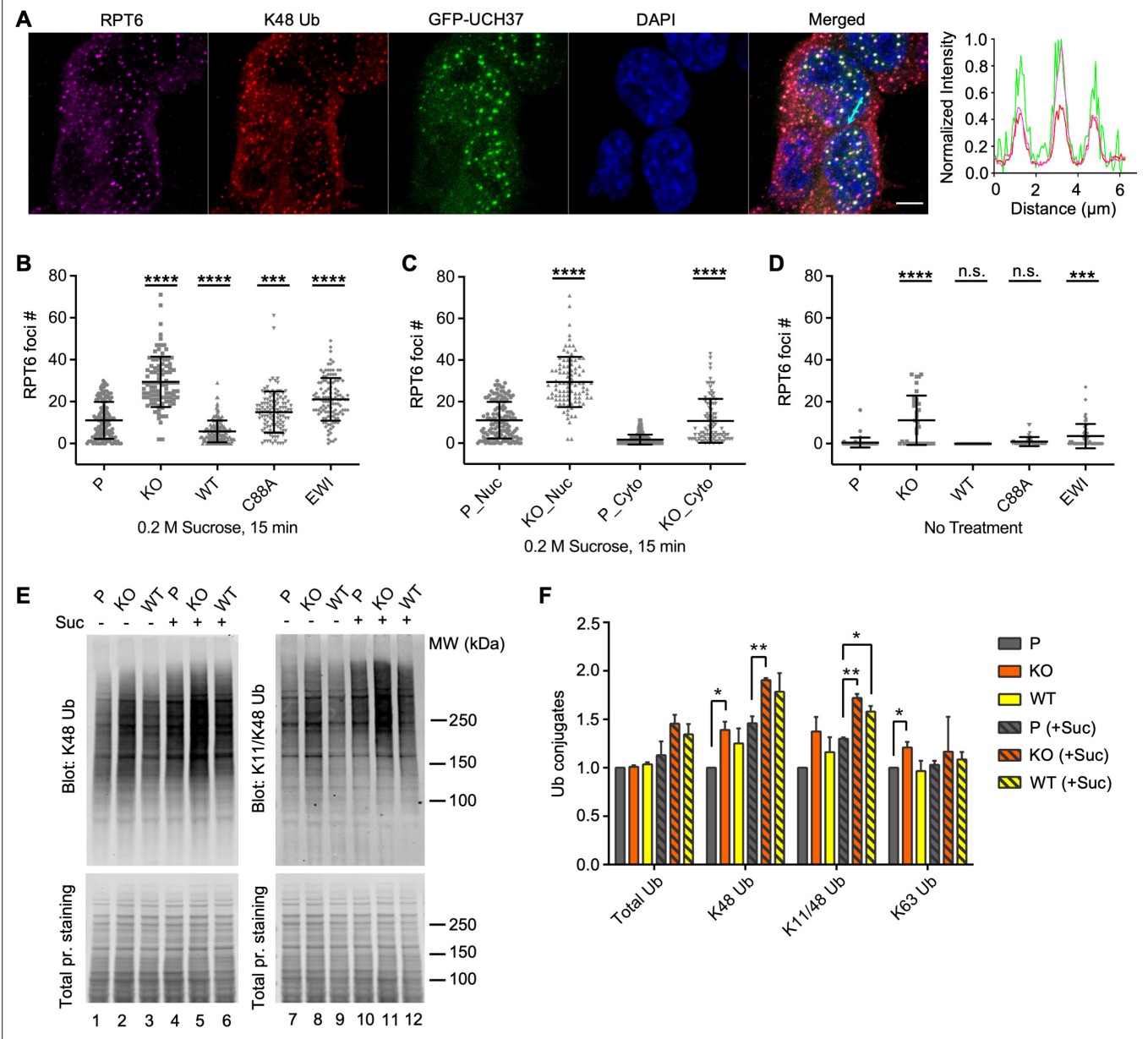

**Figure 5.** UCH37 activity regulates proteasome condensates upon osmotic stress. (**A**) HCT116 cells expressing Flag-GFP-UCH37(C88A) were treated with 0.2 M sucrose for 15 min, then fixed and immunostained with RPT6 and K48-specific antibodies. The line profile represents the magenta (RPT6), red (K48), and green (GFP) channel intensities along the arrow shown in cyan (merged panel). Scale bar, 5 μm. (**B–D**) RPT6 foci numbers in each cell were quantified and are shown as mean ± standard deviation (SD); $n > 100$ cells were measured for each cell type. Unpaired $t$-tests were performed between each cell type and P: ***, $p < 0.001$; ****, $p < 0.0001$; n.s., not significant. (**E**) Whole-cell lysates were collected from P, KO, and WT cells with or without 30 min 0.2 M sucrose treatment, and then analyzed by sodium dodecyl sulphate-polyacrylamide gel electrophoresis (SDS–PAGE) and immunblotting with the indicated antibodies. (**F**) Quantification of Ub conjugates from whole-cell lysates immunoblotted with FK2 antibody (for total Ub conjugates) or with linkage-specific anti-Ub antibodies. Representative blots are shown in (**E**). Anti-Ub signals relative to those in P were plotted after normalization using the signal intensities from total protein stain. Mean ± SD from two independent replicates are plotted. Unpaired $t$-tests were performed between each cell type and P: *, $p < 0.05$; **, $p < 0.01$.

The online version of this article includes the following source data and figure supplement(s) for figure 5:

**Source data 1.** Raw western data in *Figure 5E, F*.

**Figure supplement 1.** UCH37 regulates branched Ub chain-containing inclusions induced by proteolytic stresses.

**Figure supplement 1—source data 1.** Raw western data in supplement 1B.

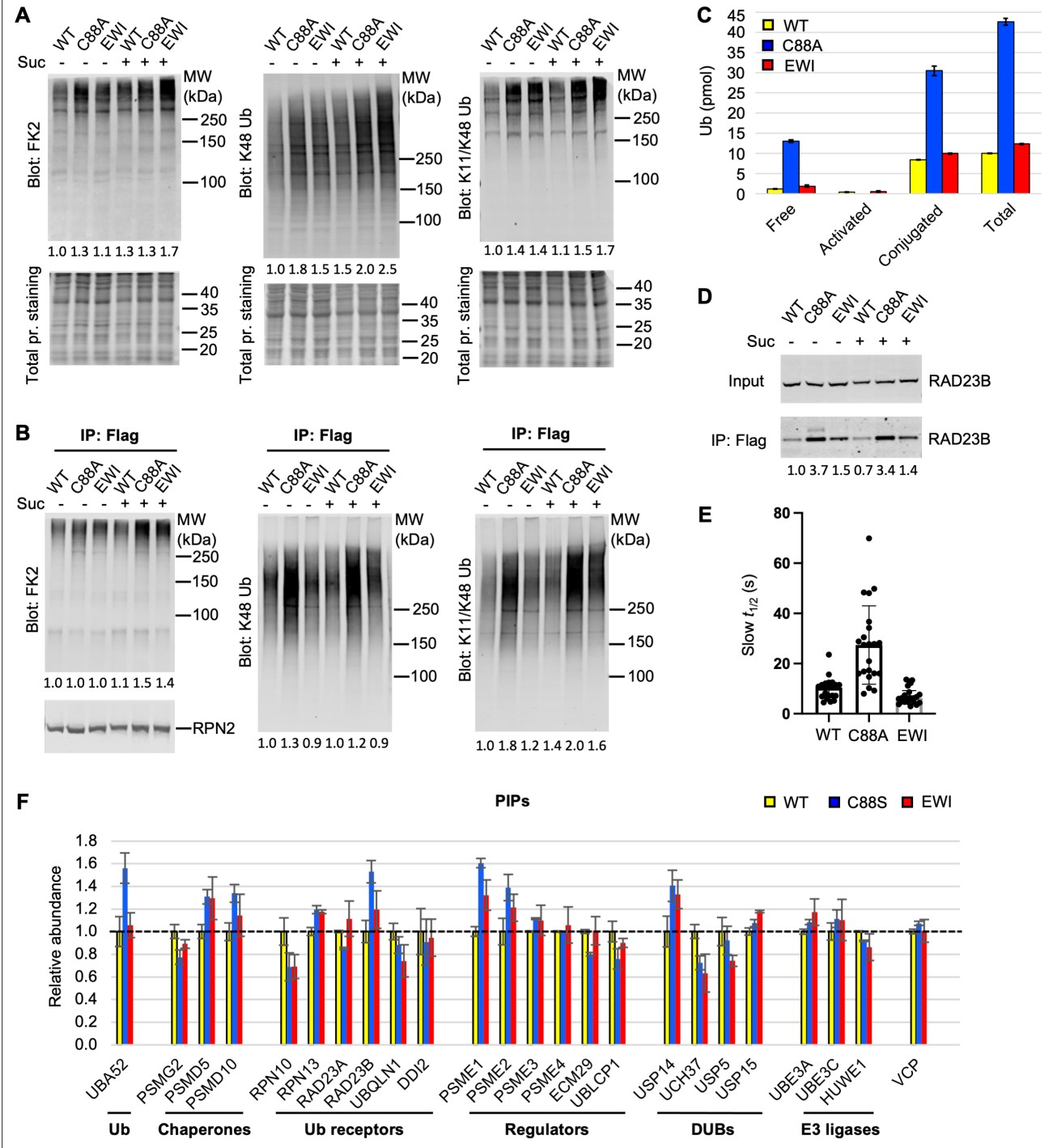

**Figure 6.** UCH37(C88A)-containing proteasomes accumulate polyUb species and RAD23B. (**A**) Soluble cell lysates were collected from WT, C88A, and EWI cells with or without 30 min 0.2 M sucrose treatment, analyzed by sodium dodecyl sulphate-polyacrylamide gel electrophoresis (SDS–PAGE) and immunblotting with indicated antibodies. Numbers indicate Ub signals relative to those in untreated WT cells after normalization against total protein stain signals. (**B**) UCH37-containing proteasomes were immunoprecipitated and analyzed as described in (**A**). Numbers below the lanes indicate Ub signals relative to those in untreated WT cells after normalization against RPN2 signals. (**C**) Different types of Ub species associated with UCH37-containing proteasomes immunoprecipitated from HEK293 cells were quantified by a free Ub sensor-based assay (*Choi et al., 2019*). Shown are mean ± SD from two independent replicates. (**D**) RAD23B accumulation detected on UCH37(C88A)-containing proteasomes as described in (**A**) and (**B**). (**E**) Fluorescence recovery after photobleaching (FRAP) analysis of GFP-UCH37 after 0.2 M sucrose treatment demonstrates that C88A-containing proteasomes are less mobile. At least 20 foci from 7 or 8 cells were analyzed from each cell line. After fitting the FRAP curve with two exponentials, the slow $t_{1/2}$ of each focus were plotted with mean and SD indicated. (**F**) UCH37-containing proteasomes from WT, C88S, and EWI cells were isolated with

*Figure 6 continued on next page*

*Figure 6 continued*

immobilized anti-GFP nanobodies, trypsin digested, and the peptides analyzed by tandem mass tag (TMT) mass spectrometry. Protein abundances were measured from two independent pulldowns, each analyzed by mass spectrometry in triplicate, and plotted as mean ± coefficient of variation (CV) after normalizing against signals from WT samples.

The online version of this article includes the following figure supplement(s) for figure 6:

**Source data 1.** Raw western data in *Figure 6A, B, D*.

**Source data 2.** Excel file of fluorescence recovery measurements for each sucrose-induced focus.

**Source data 3.** Excel file of proteomic analyses of UCH37-associated proteasomes by tandem mass tag (TMT).

**Figure supplement 1.** UCH37(C88A) and UCH37(C88S)-containing proteasomes accumulate polyUb species.

**Figure supplement 1—source data 1.** Raw western data in supplement 1A, B, C.

in the complex with monoUb described in previous crystal structures (*Vander Linden et al., 2015*; *Sahtoe et al., 2015*). These structures showed that the UCH37 UCH domain binds the S1-site Ub through extensive contacts with the classical Ub L8–I44–V70 hydrophobic patch as well as the extended C-terminal tail. These interactions are also observed for the K48-linked distal Ub of the K6/K48 $Ub_3$ in our NMR analysis. Unexpectedly, we observed additional perturbations of surface residues in the Ub α-helix, most notably K29, D32, and E34, which may represent new interactions with the S1 site that are promoted upon binding the branched chain. Although we have not identified where the second Ub-binding site is on UCH37, both our NMR analysis and mutagenesis data support that L8 and I44 of the non-K48-linked distal Ub are required for debranching. Additionally, UCH37 might also have contacts with the C-terminal tail of the K6-linked distal Ub. In contrast, debranching did not depend on the hydrophobic patch of the proximal Ub. We also observed significantly smaller perturbations of NMR signals in the proximal Ub upon binding of K6/K48 $Ub_3$ to UCH37–RPN13$^C$, suggesting that the proximal Ub serves primarily to position the distal Ub units rather than interact directly with the enzyme. It is also possible that the proximal Ub is responsible for displacing the ASCL in concert with positioning of the distal Ub units.

The affinities between the several branched $Ub_3$ chains we tested and UCH37–RPN13$^C$ are generally weak; these were either measured directly as $K_D$s or inferred from $K_M$ values. Thus, efficient debranching in vivo likely requires additional interactions that bring or retain polyUb–protein substrates onto the 26S proteasome. Full-length RPN13 contains an N-terminal PRU domain that can bind Ub (*Schreiner et al., 2008*). Although we did not observe differences in activities between UCH37–RPN13$^C$ and UCH37–RPN13$^{FL}$ with a $Ub_3$ substrate, it is possible that longer polyUb chains engage the PRU domain.

Recently, it was reported that 10–20% of polyUb chains in cells may contain branches (*Swatek et al., 2019*). With a few exceptions (*Yau et al., 2017*; *Meyer and Rape, 2014*; *Kaiho-Soma et al., 2021*; *Ohtake et al., 2018*), very little is known about the distributions of Ub–Ub linkages in branched polyUb chains, or the identities of the proteins modified by branched forms of polyUb. Upon osmotic or oxidative stresses, we observed cellular inclusions that contain proteasomes and are stained by a K11/K48-bispecific antibody, suggesting that at least a subset of proteins in those inclusions are modified with branched polyUb chains. Unfortunately, linkage-specific reagents to detect polyUb are very limited, and we were unsuccessful in our attempts to use a K6-linkage-specific anti-Ub affimer (*Michel et al., 2017*) to stain cells; thus, we have no evidence as to whether K6/K48-branched polyUb or K6-linked Ub accumulates in the stress-induced proteasome foci.

In cells lacking UCH37, the number of proteasome-containing intracellular inclusions are significantly increased, both in the absence and presence of proteolytic stress. This is accompanied by accumulation of K48 and K11/K48-linked polyUb species in the UCH37 KO cells. Overexpression of UCH37 WT not only rescues the KO phenotypes, but further reduces sucrose-induced foci in comparison with the parental cells. In contrast, overexpression of catalytically inactive UCH37, C88A or EWI, increases proteasome foci. We conclude that debranching by UCH37 promotes the dissolution of proteasome degradation centers. Interestingly, when we examined UCH37-containing proteasomes, we found that C88A, but not WT or EWI-containing proteasomes, accumulate polyUb species that include both free and conjugated Ub. Additionally, in comparison with WT or EWI, C88A proteasomes exhibit much slower association/dissociation kinetics in sucrose-induced foci. These observations suggest that, in the absence of debranching activity, UCH37 interactions with the polyUb species lead to their

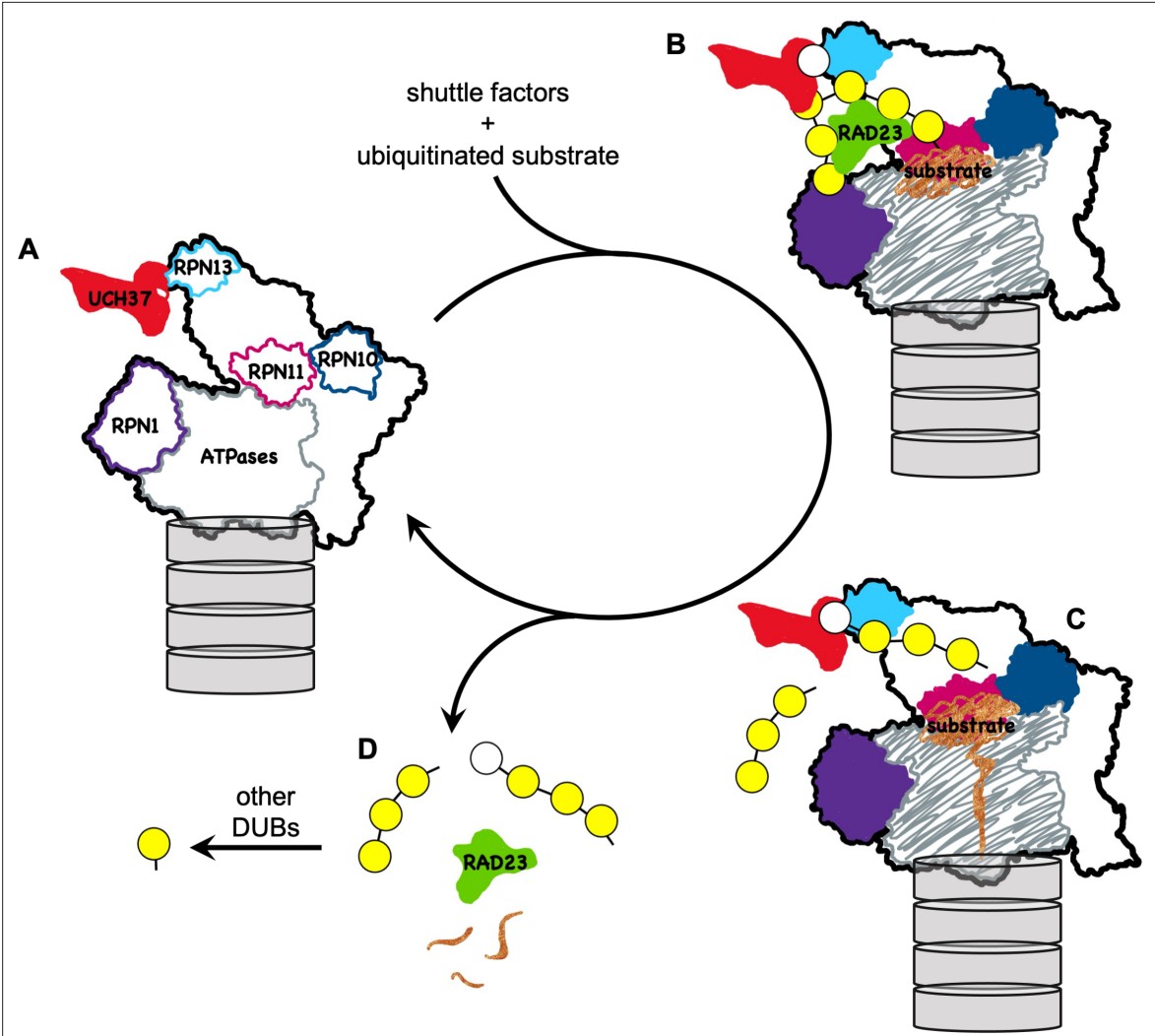

**Figure 7.** A model for how UCH37 promotes substrate processing and recycling of the proteasome through debranching of polyUb chains. (**A**) A schematic of singly capped 26S proteasome in s1 state (PDB code: 4CR2) with locations indicated for intrinsic Ub receptors (RPN1, RPN10, and RPN13) and associated DUBs (RPN11 and UCH37); for clarity, USP14 is not shown. (**B**) RAD23 delivers a substrate (*brown*) modified by polyUb with a single branch point. UCH37 binds both distal Ubs at the branch point. K48-linked Ubs are shown in *yellow* whereas a non-K48-linked Ub is shown in *white*. (**C**) A substrate-engaged proteasome (PDB code: 4CR4) where substrate starts to be translocated and polyUb is removed *en bloc* by RPN11. Whether debranching by UCH37 occurs before or after RPN11 action is unknown. (**D**) UCH37 cleavage of the K48 linkage at the branch point facilitates release of free (poly)Ub chains and the shuttle factors.

retention at the proteasome, which in turn abnormally retains other dynamic PIPs such as RAD23B and USP14.

Altogether, these results suggest that UCH37 functions to clear polyUb chains, most likely those that contain branches, from the proteasome (*Figure 7*). It has been shown that branched polyUb can promote substrate degradation by enhancing targeting to either p97/VCP or proteasomes (*Yau et al., 2017*; *Meyer and Rape, 2014*), although the underlying mechanism is unclear. With Ub$_3$ chains, we did not observe a binding preference for branched over linear chains by RAD23B as long as a K48 linkage is present, consistent with previous reports (*Varadan et al., 2005*; *Boughton et al., 2020*; *Nakasone et al., 2013*). Nevertheless, it is possible that endogenous forms of branched polyUb bind proteasomes more tightly by simultaneously engaging multiple Ub receptor sites. Proteasome-associated UCH37 could efficiently disassemble these Ub chains by severing the K48 linkage at branch points, thereby shortening the chains and also promoting chain release because most proteasomal Ub receptors prefer chains with K48 linkages (*Shi et al., 2016*; *Lu et al., 2020*; *Tsuchiya et al., 2017*). This model of UCH37 function requires that debranching is slow enough to not compete with substrate

targeting, engagement, and the initiation of degradation. For the in vitro degradation of a small model substrate such as titin-I27, substrate processing is complete in less than 1 min (*Bard et al., 2019*). Thus, with a $k_{cat}$ of 3.2 min$^{-1}$ or less, deubiquitination by UCH37 is unlikely to cause premature release of polyUb–protein substrates from the 26S proteasome. This is in contrast to USP14, another proteasome-associated DUB, which exhibits much faster kinetics and can remove the Ub signal prior to substrate commitment. In another scenario, it is also possible that binding to multiple Ub receptor sites on the proteasome slows or stalls substrate translocation; debranching by UCH37 could help to alleviate that problem. These possibilities for UCH37 function are not mutually exclusive.

## Materials and methods

### Cell culture

HCT116 cells were maintained in McCoy's 5A modified medium (Hyclone) supplemented with 10 % fetal bovine serum and 1.5 mM glutamine. Flp-In T-REx and Flp-In 293 cells were cultured in Dulbecco's modified Eagle's medium (Corning) supplemented with 10 % fetal bovine serum, 2 mM glutamine, and penicillin–streptomycin. All cell lines were kept in a humidified incubator at 37 °C with 5 % CO$_2$.

HCT116 cells are from ATCC (CCL-247). Flp-In T-REx 293 cells are from Thermo Fisher Scientific (R78007). Flp-In 293 cells are from Thermo Fisher Scientific (R75007). We did not authenticate these cell lines upon purchase. Mycoplasma testing was routinely performed using a PCR mycoplasma detection kit from Abm (G238).

### Generation of stable cell lines

UCH37-KO cells were generated by CRISPR/Cas9. A pair of gRNAs was used to introduce genomic deletions in the first exon following protocols described by *Bauer et al., 2014* Briefly, guide sequences 5′GGCATTGCCCGTCATGGCCC3′ and 5′GTCTTCACCGAGCTCATTAA3′ were cloned into pX330-U6-Chimeric_BB-CBh-hSpCas9 (a gift from Feng Zhang; Addgene plasmid #42230) and cotransfected with a GFP-expressing plasmid into HCT116 cells using Lipofectamine 2000. Forty-eight hours after transfection, single cells with high GFP expression were sorted by flow cytometry. Candidate clones were screened by PCR and successful KOs were further confirmed by western blot analysis.

Flag-GFP-tagged UCH37 (WT, C88A, C88S, and EWI) were cloned into pQCXIP (Clontech). Retroviruses were produced by cotransfection of pCI-VSVG (a gift from Garry Nolan; Addgene plasmid #1733) and pQCXIP-UCH37 plasmids into Phoenix-GP cells. Forty-eight hours after transfection, the viral supernatants were harvested, filtered, and added to UCH37 KO cells with 4 µg/ml polybrene. Infected cells were selected with 1 µg/ml puromycin and sorted subsequently by flow cytometry to enrich for low GFP-expression cells.

Flag-HA-tagged UCH37 (WT, C88A, ΔCTD, ΔCTD/C88A, EWI, and EWI/C88A) were cloned into pcDNA5/FRT/TO (Invitrogen). Stable cell lines that inducibly express these proteins were constructed from Flp-In T-REx 293 cells following manufacturer's instructions. HA-tagged UCH37 (4KR and 4KR/C88A) were cloned into pcDNA5/FRT and stable cell lines that express these proteins were constructed from Flp-In 293 cells following the manufacturer's instructions.

### Immunoblotting

Whole-cell lysates were obtained by lysing cells in 50 mM Tris–HCl, pH 7.6, 10 mM EDTA, 0.5 % SDS, and protease inhibitor cocktail (Sigma P8340), followed by sonication and clarification by centrifugation. After determining protein concentrations by the BCA assay, equal amounts of lysates were loaded and separated by SDS–PAGE using 4–12% SurePAGE Bis-Tris (GenScript) or 3–8% NuPAGE Tris-Acetate (Thermo Fisher Scientific) gels and then transferred to 0.22 µm nitrocellulose membranes. Membranes were subjected to Revert Total Protein Stain (LI-COR), followed by blocking with 5 % nonfat dry milk in Phosphate Buffered Saline-Tween 20 (PBS-T) (0.05 % Tween) for 1 hr at RT before incubation with primary antibodies at 4 °C overnight. Membranes were then washed with PBS-T three times and further incubated with IRDye secondary antibodies (LI-COR Biosciences) for 1 hr at RT. Signals were visualized with a LI-COR Odyssey CLx Imaging System and quantified using Image Studio. Primary antibodies used include: UCH37 (Abcam ab133508, 1:1000); RPN2 (Boston Biochem AP-104, 1:1000), RPN13 (custom made *Yao et al., 2006*, 1:1000); PSMB5 (Cell Signaling Technology 12,919 S, 1:1000); Ub (P4D1; Santa Cruz Biotechnology sc-8017, 1:1000); Ub conjugates (FK2; Enzo

Life Sciences PW8810, 1:1000); K48-specific Ub (Millipore 05-1307, 1:2000); K11/K48-bispecific Ub (*Newton et al., 2008*) (Genentech, 1:1500); K63-specific Ub (Millipore 05-1308, 1:1000); K6 affimer (Avacta AVA00101, 0.1 µg/ml); RAD23B (Santa Cruz Biotechnology sc-67225, 1:1000).

## Immunofluorescence and FRAP

Cells were grown on coverslips to 70–80% confluency. For proteolytic stresses, HCT116 cells were treated with 0.2 M sucrose or 0.5 mM $NaAsO_2$ for indicated times before they were fixed with 4 % paraformaldehyde in PBS for 15 min at room temperature and permeabilized with cold methanol for 10 min at −20 °C. After washing with PBS, cells were then blocked with 3 % BSA for 1 hr and incubated with primary and secondary antibodies diluted in 1 % BSA and 0.1 % Triton X-100 in PBS. Primary antibodies used include: RPT6 (Enzo Life Sciences PW9265, 1:250); K48-specific Ub (Millipore 05-1307, 1:250); RPN13 (custom made *Yao et al., 2006*, 1:100); and K11/K48-bispecific Ub (Genentech, 1:1500). Host-specific secondary antibodies conjugated with Alexa Fluor 488/594/647 dyes (Thermo Fisher Scientific) were used. Where needed, nuclei were stained with 4′,6-diamidino-2-phenylindole (DAPI) and whole cells were stained with HCS CellMask Green (Thermo Fisher Scientific) before mounting. Coverslips were mounted onto slides with ProLong Diamond Antifade (Thermo Fisher Scientific).

Images were acquired using a Zeiss LSM 880 confocal microscope with a 63× oil objective/NA1.40. All images were acquired in *Z*-stacks and maximum projections were used for quantification. Quantification of foci in images was done using CellProfiler v.3.1.9 (*McQuin et al., 2018*). DAPI signals were used to define nuclei, which were segmented by the Otsu threshold method using a two-class strategy. HCS CellMask signals were used to define cell boundary with the Watershed method. To identify RPT6 foci within the nuclei, images were segmented by the Robust Background method using a global strategy. The averaging method was set as mean and the variance set at 7 SDs. Statistical analyses were performed using GraphPad Prism v.8. Pairwise analyses used the two-tailed unpaired *t*-test; the p values are denoted as **<0.01, ***<0.001, and ****<0.0001. At least 100 cells were counted for each analysis.

FRAP experiments were performed using a Zeiss LSM 880 confocal microscope with 100× oil objective/NA1.46. Acquisition used live HCT116 cells expressing Flag-GFP-UCH37 (WT, C88A, or EWI) within 1 hr of treatment with 0.2 M sucrose, in an incubated stage chamber maintained at 37 °C with a humidified 5 % $CO_2$ atmosphere. Photobleaching of GFP-containing foci was performed with the 488 nm line from a 35 mW Ar laser operating at 100 % power. The pinhole was set to 2 µm and fluorescence recovery was monitored at intervals of 200 ms using the 488 nm laser line at 1 % power and a GAsP PMT detector. For each cell line, data were obtained from two to five foci per nucleus and seven or eight cells. Each bleached spot corresponded to a circle of 1.4 µm diameter. Normalized FRAP curves were generated from raw data after background subtraction as described (*McNally, 2008*). After normalization, the curves were fit by equations describing a two-exponential fluorescence recovery using GraphPad Prism. The fits were constrained to use a common *Fast* component while the *Slow* component for each focus' recovery was allowed to vary.

## Expression and purification of recombinant proteins

His-TEV-UCH37 (WT, C88A, or C88S) and GST-TEV-RPN13[C] were expressed in *Escherichia coli* and purified as described previously (*Vander Linden et al., 2015*). For copurifications, the harvested UCH37 and RPN13 (RPN13[C] or RPN13[FL])-expressing bacterial cell pellets were combined in a 1:1 ratio as described (*Vander Linden et al., 2015*). For NS-UCH37, the pET151-hUCH37(ISF1) plasmid (*Vander Linden et al., 2015*) was modified so that Met1 in UCH37 is replaced with an N-terminal Ser residue following cleavage by TEV protease. Full-length RPN13 was expressed from pET19b with an N-terminal 10xHis-tag in Rosetta2 BL21(DE3) cells (*Yao et al., 2006*). Log-phase cultures were shifted to 18 °C, induced with 0.4 mM IPTG, and incubation at 18 °C continued overnight before cells were harvested. His10-Rpn13[FL] was purified using HisPur Ni-NTA agarose (Thermo Fisher Scientific) following the manufacturers' instructions. The eluted proteins were further purified by gel filtration (Superdex 200 column; GE Healthcare) in PBS with 5 mM β-mercaptoethanol. Sequences of different versions of UCH37 proteins are shown in *Supplementary file 1*.

Human *RAD23B* was PCR amplified from cDNA and cloned into pGEX-6P between BamHI and XhoI sites. GST-RAD23B was expressed in BL21(DE3) Codon Plus cells grown at 37 °C to an OD600 of 0.7, and induced with 0.2 mM IPTG at 15 °C for 16 hr. Purification was achieved by affinity chromatography

(GSTrap FF 5 ml column [GE Healthcare]) in 25 mM Tris, pH 7.6, 300 mM NaCl, 2 mM dithiothreitol and ethylenediaminetetraacetic acid (DTT, 1 mM EDTA). The GST tag was removed by precision protease and RAD23B was buffer exchanged into 50 mM 4-(2-hydroxyethyl)-1-piperazineethanesulfonic acid (HEPES), pH 7.5, 150 mM NaCl, 1 mM DTT, and 10 mM EDTA.

## Synthesis of polyubiquitin chains

Ub and Ub mutants were expressed in Rosetta2 BL21(DE3) cells and purified by established procedures. E1 (*Berndsen and Wolberger, 2011*), E2 (UBCH5C *Lorick et al., 2005*, E2-25K *Haldeman et al., 1997*, CDC34 *Choi et al., 2010*, UBE2S-UBD *Bremm and Komander, 2012*, and UBC13/MMS2 *Hofmann and Pickart, 2001*, and E3 NleL *Lin et al., 2011*) enzymes were expressed and purified as described. General strategies and reaction conditions followed those described by *Raasi and Pickart, 2005*:

- $[Ub]_2–^{6,48}Ub$ was synthesized by combining Ub(K6R,K48R) with Ub(D77) or Ub(ΔGG) in a 2:1 molar ratio with 50 nM E1, 10 μM UBCH5C, and 1 μM NIeL in the ubiquitination buffer (50 mM Tris–Cl, pH 7.5, 5 mM $MgCl_2$, 50 mM NaCl, 0.5 mM EDTA, 0.5 mM DTT, 10 mM ATP, 50 mM creatine phosphate, 3 U/ml creatine kinase, 0.3 U/ml pyrophosphatase, and 5 % glycerol).
- $[Ub]_2–^{11,48}Ub$ was synthesized by first generating K11-linked $Ub_2$ with Ub(K11R,K48R), Ub(D77), or Ub(ΔGG) as described (*Bremm and Komander, 2012*). The resulting $Ub_2$ was treated with AMSH, purified by cation-exchange chromatography on a Mono S (GE Healthcare) column, then incubated with Ub(K48R), E1, and E2-25K to add the K48 branch.
- $[Ub]_2–^{48,63}Ub$ was synthesized by first generating K63-linked $Ub_2$ with Ub(K63R,K48R), Ub(D77) or Ub(ΔGG) as described (*Raasi and Pickart, 2005*). After $Ub_2$ was purified by cation exchange on a Mono S (GE Healthcare) column, it was incubated with Ub(K48R), E1, and E2-25K to add the K48 branch.
- Linear K6-, K48-, or K63-linked $Ub_3$ was built one Ub at a time as described (*Raasi and Pickart, 2005*). To build K6-linked Ub chains, the building blocks all contained the K48R mutation and NleL was used to direct the K6 linkage.
- $[Ub]_2–^{6,48}Ub$ containing Ub(L8C) as the K6-linked distal Ub was synthesized by a two-step strategy. First, K48-linked $Ub_2$ was built with Ub(K6R,K48R) and Ub(ΔGG) using E2-25K to make the K48 linkage. Then Ub(K6R,L8C,K48R) was added to the purified $Ub_2$ and NleL was used to make the K6 linkage. To carboxymethylate the L8C sidechain, 250 mM sodium iodoacetate (pH ~7.5) was added to the purified $Ub_3$ and incubated in the dark at RT for 30 min. The reaction was quenched with β-mercaptoethanol and the protein product purified over a desalting column.

For NMR studies, K6/K48-branched $Ub_3$ chains were made with domain-specific $^{15}N$ isotope labeling in a stepwise manner as detailed elsewhere (*Assfalg and Fushman, 2005*). For example, to make the $Ub_3$ with $^{15}N$-labeled K48-linked distal Ub, we started with ~10 mg each of the appropriate chain-terminating Ub mutants: $^{15}N$-labeled Ub(K6R,K48R) and unlabeled Ub(D77) in a 2 ml reaction mixture with 0.5 μM E1 and 50 μM E2-25K, protein breakdown mix (*Raasi and Pickart, 2005*), 2 mM ATP, and 3 mM TCEP for 12 hr at 30 °C. The K48-linked $Ub_2$ product was purified by cation-exchange chromatography (5 ml HiTrap SP HP column; GE LifeSciences) using a gradient of NaCl in 50 mM ammonium acetate, pH 4.5, and then buffer exchanged into 50 mM Tris, pH 8.0 for the next step of the synthesis. The $^{15}N$-labeled K48-linked $Ub_2$ was mixed with unlabeled Ub(K6R,K48R) at 1:1 molar ratio in a 2 ml reaction containing 0.5 μM E1, 50 μM UbcH7 (E2), and 50 μM NleL (E3) in the protein breakdown mix, 2 mM ATP, 3 mM TCEP for 12 hr at 30 °C. Cation-exchange chromatography yielded pure K6/K48-linked branched $Ub_3$. Finally, the protein was exchanged into 20 mM sodium phosphate buffer, pH 7.2, containing 100 mM NaCl, 1 mM TCEP, 1 mM EDTA, and 0.2 % (wt/vol) $NaN_3$. The masses of the $Ub_2$ and $Ub_3$ products were verified by SDS–PAGE and ESI-TOF mass spectrometry. A similar procedure was used to assemble K6/K48-linked branched $Ub_3$ chains with $^{15}N$-labeled K6-linked distal or proximal Ub.

## Synthesis of crosslinked branched Ub chain mimics

1,3-Dichloroacetone (DCA)-based crosslinking was performed essentially as described (*Long et al., 2014*). For Mimic1, His6-Ub(G76C) and Ub-$^{48}$Ub(K6C) were mixed at equal molar ratio in 50 mM sodium tetraborate, pH 8.5. 5 mM TCEP was added at RT for 30 min followed by DCA at an amount equal to one-half of the total sulfhydryl groups in the reaction. After 30 min on ice, the reaction was

quenched with 10 mM β-mercaptoethanol. Branched Ub$_3$ was purified by gel filtration on Superdex 75 (GE Healthcare). For Mimic2, His6-Ub$_{75}$-mercaptoethylamide was prepared using intein chemistry modified from a previous work (*Wilkinson et al., 2005*). Briefly, His6-Ub$_{75}$ (WT or L8A,I44A)-intein-CBD were expressed from pTYB2 plasmids in ER2566 *E. coli* (New England Biolabs). The expressed fusion proteins were purified on chitin resin and Ub was released by incubating the resin in 100 mM cysteamine at 4 °C overnight. The resulting His6-Ub$_{75}$-mercaptoethylamide was further purified on Ni-NTA agarose to remove excess cysteamine. The subsequent crosslinking reaction was performed as described for Mimic1.

## Deubiquitination assays

Purified UCH37 and RPN13$^C$ were mixed at equal molar concentrations and incubated at RT for 15 min before addition into assay buffer (50 mM HEPES, pH 7.5, 50 mM NaCl, and 2 mM DTT). For gel-based assays, polyubiquitin substrates were added and reactions were incubated at 37 °C with aliquots taken at indicated time points for analysis by SDS–PAGE. Ub-AMC hydrolysis assays were performed as described (*Vander Linden et al., 2015*).

The kinetics of debranching by UCH37–RPN13$^C$ were determined by real-time monitoring of product release with the free Ub sensor, Atto532-tUI (*Choi et al., 2019*). Each reaction contained 500 nM Atto532-tUI, 20 or 40 nM preformed UCH37–RPN13$^C$ complex, and 0–250 µM substrates in assay buffer (50 mM HEPES, pH 7.5, 5 mM DTT, 50 mM NaCl, and 0.2 mg/ml ovalbumin). For reactions containing 50, 100, or 250 µM substrate, Ub sensor concentration was increased to 1.5 µM. Reactions at 30 °C were initiated by enzyme addition and fluorescence was monitored over 500 s using a FluoroMax-4 Spectrofluorometer (HORIBA Scientific) with 532 nm excitation (slit width 4 nm) and 553 nm emission (slit width 3 nm). A standard curve was created with known amounts of Ub in order to convert fluorescence increase to free Ub concentration. $V_0$ was calculated from initial linear portion of the progress curve. All reactions were done in duplicate. $K_M$ and $k_{cat}$ values were determined by nonlinear fits to Michaelis–Menten kinetics using GraphPad Prism.

## Microscale thermophoresis (MST)

MST assays were performed in a Monolith NT.115 using Standard Treated Glass Capillaries (NanoTemper Technologies MO-K002). His-TEV-UCH37(C88S)–RPN13$^C$ was labeled with Monolith His-Tag labeling Kit RED-tris-NTA (MO-L008) following the manufacturer's instructions. Full-length RAD23B was labeled with the Monolith Protein Labeling kit RED-NHS 2nd Gen (MO-L011). Each assay contained 50 nM His-TEV-UCH37(C88S)–RPN13$^C$, or 20 nM RAD23B, and varied polyUb concentrations. The assays were performed in PBS containing 0.05 % Tween-20, at 80 or 100 % of excitation power, and 40 % of MST power. MST traces were analyzed at 5 s. 1 mM DTT was supplemented to assays with RAD23B. Change of fluorescence, $F_{norm}$‰, was plotted against polyUb concentration and the curves were fitted with a single-site-binding model to determine the binding affinity ($K_D$) using GraphPad Prism.

## Immunoprecipitation

Cells from one 10 cm dish were washed with PBS, harvested, and resuspended in 350 µl low salt lysis buffer (20 mM HEPES, 50 mM NaCl, 10 mM KCl, 1.5 mM MgCl$_2$, 10 % glycerol, 0.5 % Triton X-100, and pH 7.9) supplemented with 10 mM iodoacetamide and protease inhibitors (Sigma P8340) on ice for 30 min. Lysates were cleared by centrifugation and incubated with 30 µl anti-Flag agarose (Sigma A2220) at 4 °C overnight with rotation. Beads were washed three times with lysis buffer and bound proteins were eluted with 2× Laemmli sample buffer and analyzed by immunoblotting. Alternatively, to quantify the (poly)Ub species, bound proteins were eluted with 0.2 mg/ml 3xFlag peptide. Total, free, activated, and conjugated Ub species were determined using Atto532-labeled tUI following a previously described protocol (*Choi et al., 2019*).

## Mass spectrometry analysis of UCH37-containing proteasomes

Native proteasomes were purified according to *Wang et al., 2007*. Briefly, HCT116 cells from 5× 15 cm dishes were washed once with PBS and crosslinked in 0.025 % formaldehyde in PBS for 10 min at 37 °C, followed by the addition of 0.125 M glycine to quench the crosslinker. After washes, cells were harvested and lysed in lysis buffer (50 mM NaPi, pH 7.5, 100 mM NaCl, 10 % glycerol, 0.5% NP-40,

5 mM MgCl$_2$, 5 mM ATP, and 1 mM DTT) supplemented with protease inhibitors (Sigma P8340) and phosphatase inhibitors (GoldBio GB-450). Lysates were passed through a 21 G needle 20 times, incubated on ice for 15 min, then centrifuged at 18,000 × *g* for 15 min at 4 °C. The supernatant was incubated with anti-GFP nanobody crosslinked to Sepharose (*Schellenberg et al., 2018*) for 2 hr at 4 °C with rotation. The beads were washed twice with lysis buffer and once with wash buffer (25 mM NaPi, pH 7.5, 150 mM NaCl, 5 mM ATP, 5 % glycerol), then resuspended in 25 mM ammonium bicarbonate, 5 mM TCEP, 8 M urea, 10 mM iodoacetamide, and incubated in the dark at RT for 30 min. After dilution with 25 mM ammonium bicarbonate, on-bead digestions were done sequentially with 2 µg LysC (Wako chemicals 125-05061) in 4 M urea and 4 µg trypsin (Promega ADV5113) in 1.2 M urea before quenching with 1 % formic acid. Peptides were collected and combined with multiple washes of the beads with 0.1 % formic acid in 25 % acetonitrile (ACN).

For TMT labeling, individual digested peptide mixtures were cleaned using Waters C18 Sep-PAK cartridges and vacuum concentrated. They were then diluted using 50 mM triethyl ammonium bicarbonate (TEAB) and adjusted to pH ~8 through multiple cycles of dilution with water and TEAB and vacuum concentration. The final 50 µl mixtures were individually labeled using 20 µg of a single channel of TMT10plex isobaric labeling reagent (Thermo Fisher Scientific PI90110) in anhydrous ACN and incubated for 1 hr at RT. Hydroxylamine was added to each sample to a final concentration of 0.25 % and incubated for 15 min with occasional vortexing to quench the labeling reaction. Samples were cleaned and desalted again using Waters C18 Sep-PAK cartridges to remove excess TMT labeling reagent and vacuum concentrated. All 10 channels of TMT10plex-labeled samples were combined and analyzed in triplicate by LC–MS/MS utilizing a Thermo Scientific EASY-nLC 1000 UPLC system coupled online to a Thermo Scientific Orbitrap Fusion Lumos Mass Spectrometer. A Thermo Scientific EASY-Spray source with a 25 cm × 75 µm PepMap EASY-Spray Column was used to separate peptides over a 90 min gradient of 6% to 35% ACN in 0.1 % formic acid at a flow rate of 300 nl/min. MS (*Deol et al., 2020*) and MS (*Lee et al., 2011*) scans were both acquired in the Orbitrap. For MS1 scans, the scan range was set from 375 to 1500 *m/z*, resolution set to 12,000, and the AGC target set to 1 × 10$^6$. For MS (*Lee et al., 2011*) scans, the resolution was set to 50,000, the AGC target was set to 1 × 10$^5$, the precursor isolation width was 0.8 *m/z*, and the maximum injection time was 110 ms. The HCD MS/MS normalized collision energy was set to 38 %.

Thermo Scientific Proteome Discoverer 2.3 software with SEQUEST was used for protein identification against a database containing all SwissProt entries for *Homo sapiens* (February 2020). Searches were performed using a 10 ppm precursor ion tolerance and the product ion tolerance was set to 0.1 Da. TMT tags on lysine residues and peptide N termini (+229.163 Da) and carboxyamidomethylation of cysteine residues (+57.021 Da) were set as static modifications, whereas oxidation of methionine residues (+15.995 Da) was set as a variable modification. Peptide-spectrum matches (PSMs) and protein false discovery rates were set as 1 %. Reporter ion intensities were adjusted to correct for the isotopic impurities of the different TMT reagents according to the manufacturer's specifications.

To compare UCH37 (WT, C88S, or EWI)-containing proteasomes, the abundance of each identified protein was first normalized against those in the WT sample. We then used the average abundance of the 19S complex subunits as a measure of the amount of proteasomes in each sample. This allowed us to compare the relative quantities of proteasome subunits and PIPs on a per-proteasome basis.

## NMR experiments and CSP mapping

All samples for NMR measurements were prepared in 20 mM sodium phosphate buffer with pH7.2 containing 100 mM NaCl, 1 mM TCEP, 1 mM EDTA, 0.2 % NaN$_3$, and 10 % D$_2$O. The NMR measurements were performed at 23 , 30 , and 37 °C on an Avance III 600 MHz Bruker NMR spectrometer equipped with a cryoprobe. The data were processed using Topspin 3.6.3 (Bruker) and analyzed using Sparky 3.114 (*Goddard and Kneller, 2002*).

Binding studies by NMR were done by adding precalculated amounts of unlabeled copurified UCH37(C88A)–RPN13$^C$ to $^{15}$N-labeled Ub$_3$ or Ub$_1$ up to ~1.2:1 molar ratio and monitoring changes in 2D $^1$H–$^{15}$N SOFAST-HMQC spectra recorded at every titration point. The starting concentration of Ub$_3$ or Ub$_1$ was 100 µM, the UCH37–RPN13$^C$ stock concentration was 268 µM.

Changes in amide peak positions in $^1$H–$^{15}$N NMR spectra were quantified as CSPs using the following equation: CSP = $[(\Delta\delta_H)^2 + (\Delta\delta_N/5)^2]^{1/2}$, where $\Delta\delta_H$ and $\Delta\delta_N$ are the chemical shift differences for $^1$H and $^{15}$N, respectively, for a given residue between the free protein and upon addition of UCH37–RPN13$^C$.

Signal attenuations were quantified as the ratio of NMR signal intensities, $I/I_0$, in the spectra measured upon ($I$) and prior to ($I_0$) addition of the ligand. To compensate for the overall reduction of the NMR signals caused by increased size (slower tumbling) of $Ub_3$ (26 kDa) upon complex formation with UCH37–RPN13$^C$ (52 kDa), the final spectra at the end of titration were recorded with a higher number of scans (see *Figure 3—figure supplement 1*), and the resulting intensities ($I$ and $I_0$) were divided by the respective number of scans. The $I/I_0$ ratio was further corrected by the volume dilution factor.

## Acknowledgements

We thank Melanine Furgason and Andrew Roddam for the initial development of the Mimic2 Ub conjugation strategy, to Joseph P Thelen for creating the UCH37 knockout cell line, and to Yun-Seok Choi for materials and advice for the real-time deubiquitination assay with tUI. This work was supported by NIH grants R01 GM098401 to TY, R01 GM115997 and R21 GM135818 to REC, R01 GM074830 and R01 GM130144 to LH, and R01 GM065334 to DF.

## Additional information

### Competing interests

Christopher P Hill: Reviewing editor, eLife. The other authors declare that no competing interests exist.

### Funding

| Funder | Grant reference number | Author |
|---|---|---|
| National Institute of General Medical Sciences | R01 GM115997 | Robert E Cohen |
| National Institute of General Medical Sciences | R21 GM135818 | Robert E Cohen |
| National Institute of General Medical Sciences | R01 GM065334 | David Fushman |
| National Institute of General Medical Sciences | R01 GM074830 | Lan Huang |
| National Institute of General Medical Sciences | R01 GM130144 | Lan Huang |
| National Institute of General Medical Sciences | R01 GM098401 | Tingting Yao |

The funders had no role in study design, data collection, and interpretation, or the decision to submit the work for publication.

### Author contributions

Aixin Song, Data curation, Formal analysis, Methodology, Project administration, Writing – original draft, Writing – review and editing; Zachary Hazlett, Justin Curtiss, Data curation, Formal analysis, Methodology, Writing – review and editing; Dulith Abeykoon, Jeremy Dortch, Sarah Bollinger Martinez, Data curation, Formal analysis, Methodology; Andrew Dillon, Data curation, Formal analysis, Writing – original draft, Writing – review and editing; Christopher P Hill, Conceptualization, Investigation, Writing – review and editing; Clinton Yu, Data curation, Formal analysis, Methodology, Writing – original draft; Lan Huang, Conceptualization, Data curation, Formal analysis, Funding acquisition, Methodology; David Fushman, Conceptualization, Data curation, Formal analysis, Funding acquisition, Investigation, Methodology, Writing – original draft, Writing – review and editing; Robert E Cohen, Conceptualization, Funding acquisition, Investigation, Methodology, Project administration, Writing – original draft, Writing – review and editing; Tingting Yao, Conceptualization, Data curation, Formal analysis, Funding acquisition, Investigation, Methodology, Project administration, Writing – original draft, Writing – review and editing

### Author ORCIDs

Aixin Song (ID) http://orcid.org/0000-0002-4377-7528
Justin Curtiss (ID) http://orcid.org/0000-0002-3996-2659
Christopher P Hill (ID) http://orcid.org/0000-0001-6796-7740
Robert E Cohen (ID) http://orcid.org/0000-0001-7397-4138
Tingting Yao (ID) http://orcid.org/0000-0003-4101-9691

### Decision letter and Author response

Decision letter https://doi.org/10.7554/eLife.72798.sa1
Author response https://doi.org/10.7554/eLife.72798.sa2

---

## Additional files

### Supplementary files

• Supplementary file 1. Sequence alignment for UCH37 proteins used in this work. The active site C88 is highlighted in yellow and EWI residues are indicated in orange.

• Transparent reporting form

### Data availability

All data contributed to the results in this manuscript are included in the manuscript and supporting files.

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
