## [Editor Report]

This study nicely addresses the role of a deubiquitylating enzyme (UCH37) in facilitating proteasomal clearance of branched polyubiquitylated substrates. Using a wide-range of chemical biological, biophysical and cell biological techniques, the authors convincingly demonstrate recognition of branched ubiquitin trimers and further demonstrate that mutations of the deubiquitylating enzyme lead to the formation of proteasomal foci in cells that are rich in polyubiquitinated species, presumably due to the loss of debranching activity. Overall, this excellent study adds to our understanding of UCH37 function, especially with regard to the newly observed phenomenon of reversible proteasome aggregation in cells.

---

## [Decision Letter]

**Decision letter after peer review:**

Thank you for submitting your article "Branched ubiquitin chain binding and deubiquitination by UCH37 facilitate proteasome clearance of stress-induced inclusions" for consideration by *eLife*. Your article has been reviewed by 3 peer reviewers, and the evaluation has been overseen by a Reviewing Editor and Philip Cole as the Senior Editor. The reviewers have opted to remain anonymous.

Recommendations for the authors:

The major conclusions of the manuscript are supported by the data but with some technical concerns/questions, as raised below.

1) Figure 1: there seems to be a bit of variability in terms of how Ub3 presents on the gels. A smear with lower bands in A, a smear with upper bands in B and a more prominent single band in D. It'd be good to include molecular weight markers for all these gels if possible. Is there any rationale for why the Ub3 is so variable? Also it's not clear why the activity against 48/63 chains is not higher given the Mol Cell manuscript by Deol et al.

2) In Figure 1D, why is RPN13-C not apparent for the left panel with Ub2-6,48Ub? RPN13-C includes the DEUBAD domain but which amino acids exactly does it span? It's also at low abundance compared to UCH37 in the right panel and in addition the level of UCH37 is much greater in the Ub-48Ub-48Ub experiment compared to 6,48. Also why does UCH37 seem to migrate a little higher for the amino acid substituted forms and is there some proteolysis for the DD mutant, as there is a lower band above the Ub3? Ub1 seems like it might not be as reliable to quantify as Ub2 for 1D; is it possible to include in 1E the plot for Ub2? One reason for raising this possibility is that there seems to be an increase in Ub2 for UCH37 alone with the DD mutant for Ub2-6,48Ub when examining Figure 1D. Are the data in Figure 1E plotted for the same UCH37 protein – ratio for DD plotted compared to DD without RPN13?

3) Figure 3: The L8A/I44A mutations in proximal Ub leads to loss of spectral effects with UCH37-RPN13C addition. This finding is interesting because it suggests that interaction with proximal Ub is not required for the debranching activity, particularly when combined with the data in Figure 3E, as noted by the authors. The text seems to also imply that the spectral effect is caused by lost intraUb interactions, but these mutations could also impact possible intermolecular interactions. Perhaps the text could be clarified a bit on this point. Also again, in Figure 3E RPN13C is barely observable – perhaps a 1D NMR spectrum can be used to resolve the abundance of RPN13C for these experiments or alternatively mass spectrometry. Mass spectrometry may also reveal why Ub3 varies.

4) Line 194: The reference to Figure 6 – supplement 1A for EWI mutant having reduced affinity for Ub seems incorrect.

5) Figure 5A is convincing but a line profile would be further helpful and the distance seems to be missing for the scale bar. These suggestions apply also to Figure 5, S1C/D.

6) A caveat of the experiments involving the C88A mutant is that it itself is ubiquitinated (Figure 6, supple 1A, right middle panel lane 1 and Figure 5, supple 1B). Does increased ubiquitin at the proteasome with the C88A mutant represent at least in part ubiquitinated UCH37 mutant for example? How would the FRAP experiments be influenced if ubiquitin on UCH37 were interacting with the proteasome (Figure 6E)?

7) "Proteasomes containing UCH37(C88A), which is catalytically inactive, aberrantly retain polyubiquitinated species as well as the RAD23B substrate shuttle factor, suggesting a defect in recycling of the proteasome." Please change to "recycling by the proteasome." I suspect the authors are not referring to the recycling of proteasomes themselves.

8) "In recent years, significant progress has been made in our understanding of how RPN11 and USP14 are regulated to coordinate with substrate processing on the 26S proteasome, but the role of UCH37 remains poorly understood."

And

"how UCH37 contributes to the functions of the proteasome or INO80 complex remains unknown."

We would suggest amending these two statements based on Strieter's Molecular Cell paper in which they showed that global proteolysis is supported by UCH37 debranching activity. It may be useful to focus instead on missing aspects from the previous study such as the mode of UCH37-binding to branched Ub and the overall role of UCH37 in proteasomal function.

9) "We noticed that the kcat we measured for K6/K48 Ub3 debranching at 30 {degree sign}C is slower than what was reported by others, which varied from 3 to 12‐fold faster at 37 {degree sign}C, whereas the KM values are similar"

The authors should note that two entirely different techniques (MS vs probe binding) and temperatures were employed in the kinetics assays, which may underlie differences in observed kinetics. One question for the authors is that given the fact that UCH37 must operate at 37 degrees in humans, where questions of its kinetics competing with proteasomal degradation may reasonably be raised, why did they choose to measure kinetics at 30 degrees? How does this relate to the argument in the discussion that relatively "slow" debranching would not compete with substrate engagement by the proteasome? Would the authors expect this to also be accurate at 37 degrees where UCH37 may be faster as per Strieter et al.?

10) While the biophysical measurements were crystal clear, aspects of the manuscript dealing with proteasomal foci formation and especially RAD23B accumulation were unclear as to what precise conclusion was arrived at by the authors. RAD23B did not appear to show any binding preference in their assays for example and they did not speculate further. These experiments almost felt like an attempt to distinguish the current manuscript from previous literature that had already demonstrated impaired proteasomal function due to UCH37 knockouts. The authors should clarify in the text whether they envision that foci are formed because of the accumulation of polyubiquitylated substrates, or, if foci formation occurs first, which then leads to the accumulation of polyubiquitylated substrates. If it is the former, isn't foci formation simply one of the many physiological outcomes from a blocked proteasome that is still recruiting polyubiquitylated species which in turn recruit more proteasomes? If there is something unique about the clustering of proteasomes in this manner, the authors should definitely highlight this unique aspect of UCH-37 function.

---

## [Author Response]

Recommendations for the authors:The major conclusions of the manuscript are supported by the data but with some technical concerns/questions, as raised below.1) Figure 1: there seems to be a bit of variability in terms of how Ub3 presents on the gels. A smear with lower bands in A, a smear with upper bands in B and a more prominent single band in D. It'd be good to include molecular weight markers for all these gels if possible. Is there any rationale for why the Ub3 is so variable? Also it's not clear why the activity against 48/63 chains is not higher given the Mol Cell manuscript by Deol et al.

Early on, we were also puzzled by the variable appearance of Ub3 band on SDS-PAGE. We found that the cause is incomplete denaturation of Ub3 by the SDS sample buffer. We prefer to avoid heating these samples prior to SDS-PAGE because heat causes aggregation of polyUb chains, a well-known phenomenon in the field. As a result, Ub3 is incompletely denatured, and its appearance on the gel depends on how long the sample had been in SDS-sample buffer. However, we have made sure that the appearance does not affect quantitation of the bands after Coomassie staining. Molecular weight markers are presented in Source Data 1.

For 48/63 branched Ub3, we have not observed robust debranching activity from UCH37-RPN13. We have used linkage-specific DUBs (Otub1 and AMSH) to confirm that the chains we synthesized have correct linkages. We have also tested different batches of 48/63 branched Ub3 made in different labs and obtained consistent results.

In the Mol Cell paper by Deol et al., the only experiment shown with 48/63 branched trimer is in Figure 1B. However, that experiment was done with TEC chains that contain non-native Ub-Ub linkages. It is also not possible to compare initial velocities using the results presented in that gel. The rest of the Deol et al., results that employed 48/63 branched chains used complex HMW (high molecular weight) chain mixtures. Due to the heterogeneity of those substrates, it is difficult to derive kinetic parameters (i.e., initial velocity, Km or Kcat values) for meaningful comparison with our results. Overall, however, it appears that debranching of the HMW 48/63 chains is slower than 6/48 or 11/48 chains (see Figures 2 and S5 in Deol et al.,), a trend that agrees with our results using homogeneous Ub3 substrates.

2) In Figure 1D, why is RPN13-C not apparent for the left panel with Ub2-6,48Ub? RPN13-C includes the DEUBAD domain but which amino acids exactly does it span? It's also at low abundance compared to UCH37 in the right panel and in addition the level of UCH37 is much greater in the Ub-48Ub-48Ub experiment compared to 6,48. Also why does UCH37 seem to migrate a little higher for the amino acid substituted forms and is there some proteolysis for the DD mutant, as there is a lower band above the Ub3? Ub1 seems like it might not be as reliable to quantify as Ub2 for 1D; is it possible to include in 1E the plot for Ub2? One reason for raising this possibility is that there seems to be an increase in Ub2 for UCH37 alone with the DD mutant for Ub2-6,48Ub when examining Figure 1D. Are the data in Figure 1E plotted for the same UCH37 protein – ratio for DD plotted compared to DD without RPN13?

We apologize that we did not define RPN13C clearly. This is the same construct as was used in Vanderlinden et al., (Mol Cell, 57:901); it encompasses aa 285-407. We have added that info in the revised manuscript, line 75.

Possibly because of its small size, RPN13C stains poorly with Coomassie. In Figure 1D, the enzyme (UCH37-Rpn13C) was used at both 0.5 μm (left panel) and 10 μm (right panel). This is because it has much higher activity against branched chains. Given the 20-fold difference in the enzyme amount, you cannot see RPN13C in the left panel. Purified RPN13C is quantified by absorbance at 280 nm. We think the quantification is reliable. In Figure 2-supplement 1C, we compared RPN13C added in trans (Lane 1) or co-purified with UCH37 (Lane 2). Because of the co-purification protocol, RPN13C that’s co-purified with UCH37 should be at 1:1 molar ratio. Comparing Lanes 1 and 2, the RPN13C bands gave rise to similar staining intensities despite that they were purified and quantified by different procedures.

The UCH37 AA and DD mutants do migrate slightly slower than the WT enzyme. This is likely due to loss of hydrophobicity provided by M148,F149 residues (and, as a result, less SDS is complexed). The DD mutant does appear to have a slight contamination by a proteolytic fragment. However, we see the same results with AA and DD mutants. The comparison in Figure 1E is made for the same enzyme with and without RPN13C. We have quantified Ub2 as the reviewer requested. The results are the same as with the Ub1 quantitation. In order to avoid overcrowding of the original figure, we put the Ub2 quantitation results in Figure 1-supplement 1D.

3) Figure 3: The L8A/I44A mutations in proximal Ub leads to loss of spectral effects with UCH37-RPN13C addition. This finding is interesting because it suggests that interaction with proximal Ub is not required for the debranching activity, particularly when combined with the data in Figure 3E, as noted by the authors. The text seems to also imply that the spectral effect is caused by lost intraUb interactions, but these mutations could also impact possible intermolecular interactions. Perhaps the text could be clarified a bit on this point. Also again, in Figure 3E RPN13C is barely observable – perhaps a 1D NMR spectrum can be used to resolve the abundance of RPN13C for these experiments or alternatively mass spectrometry. Mass spectrometry may also reveal why Ub3 varies.

The reviewer is correct that we cannot differentiate intra-Ub3 vs Ub3-UCH37 interactions from the NMR experiments. This is why the activity assay in Figure 3E is important. We have now clarified this point in Line 155. The added reference 32 also has extensive discussion on the complications in interpretation of NMR data in a similar case. We thank the reviewer for the suggestion.

As we described above (point #2), RPN13C is stained poorly. We have adjusted the contrast for Figure 3E. to make the faint bands more visible.

4) Line 194: The reference to Figure 6 – supplement 1A for EWI mutant having reduced affinity for Ub seems incorrect.

We have added the reference to Sahtoe et al., (Mol Cell 57:887)(Line 206, revised). This paper has a detailed discussion on the important contribution of Ile216 to Ub binding by UCH37. We did not purify and characterize the UCH37-EWI mutant protein. The EWI triple mutation is expected to severely reduce Ub binding, thereby making Ub-AMC assays and Km determinations impractical. We instead prioritized characterization of the EWI mutation in mammalian cells. Figure 6-supplement 1A shows that both the EWI and EWI/C88A mutants pull down much less polyUb than with WT and C88A UCH37. These results support the prediction of weakened Ub binding.

5) Figure 5A is convincing but a line profile would be further helpful and the distance seems to be missing for the scale bar. These suggestions apply also to Figure 5, S1C/D.

We thank the reviewer for the suggestion. We have added line profiles in Figure 5A, S1C/D, and we have specified the size of the scale bar in the figure legend.

6) A caveat of the experiments involving the C88A mutant is that it itself is ubiquitinated (Figure 6, supple 1A, right middle panel lane 1 and Figure 5, supple 1B). Does increased ubiquitin at the proteasome with the C88A mutant represent at least in part ubiquitinated UCH37 mutant for example? How would the FRAP experiments be influenced if ubiquitin on UCH37 were interacting with the proteasome (Figure 6E)?

We don’t think that ubiquitination of UCH37-C88A itself contributes significantly to the total ubiquitin species associated with the proteasome. In Figure 6-Supplement 1A, Lane 7, ubiquitinated UCH37-C88A runs between 50 kDa and 75 kDa. From the anti-Ub blot, it is clear that those bands are minor in comparison with the high molecular weight Ub conjugates. In addition, we present new data (Figure 6-Supplement 1B) to show that increased polyUb species are still observed in C88A-containing proteasomes even when most of the ubiquitination on C88A is blocked. We had previously mapped the ubiquitination sites on UCH37-C88A and, by mutating the major lysine ubiquitination sites to arginines (4KR), we could eliminate most UCH37 ubiquitination. Comparing immunoprecipitates from HEK293 cells expressing WT-4KR or C88A-4KR shows that similar patterns of Ub conjugates are obtained regardless of the 4KR mutation. These new results are described in Line 252. Currently, we don’t have a GFP-tagged UCH37(4KR) or the stable cell lines necessary for FRAP experiments. Nevertheless, based on the co-immunoprecipitation experiment results, we predict that the behavior would be similar to what we report in Figure 6E.

7) “Proteasomes containing UCH37(C88A), which is catalytically inactive, aberrantly retain polyubiquitinated species as well as the RAD23B substrate shuttle factor, suggesting a defect in recycling of the proteasome.” Please change to “recycling by the proteasome.” I suspect the authors are not referring to the recycling of proteasomes themselves.

We mean “recycling” in the sense of allowing an enzyme to engage in multiple rounds of catalysis. In this case, either substrates or products (probably both based on our results) cannot be efficiently released from the proteasome, thus preventing “recycling of the proteasome”. Although we believe that “recycling” is appropriate here, we are open to suggestions for a better word.

8) “In recent years, significant progress has been made in our understanding of how RPN11 and USP14 are regulated to coordinate with substrate processing on the 26S proteasome, but the role of UCH37 remains poorly understood.”And“how UCH37 contributes to the functions of the proteasome or INO80 complex remains unknown.”We would suggest amending these two statements based on Strieter’s Molecular Cell paper in which they showed that global proteolysis is supported by UCH37 debranching activity. It may be useful to focus instead on missing aspects from the previous study such as the mode of UCH37-binding to branched Ub and the overall role of UCH37 in proteasomal function.

We agree that the Strieter group’s Mol Cell paper was a major step forward in understanding UCH37. However, many mechanistic questions regarding how UCH37 activity is coordinated with substrate processing remain unanswered. Nevertheless, we have changed the aforementioned sentence to “In recent years, significant progress has been made in our understanding of how RPN11 and USP14 are regulated to coordinate with substrate processing on the 26S proteasome, a role of UCH37 was discovered only recently (ref to Deol et al.).” (Line 16)

Regarding the 2^nd^ part of this comment, what we meant was that the current structures of UCH37 do little to inform us about the function of UCH37 in either complex. Therefore, we have changed the sentence to “Despite that these structures provided clear understanding of how UCH37 activity is controlled in different contexts, they did not reveal how UCH37 contributes to the functions of the proteasome or INO80 complex.” (Line 43).

9) “We noticed that the kcat we measured for K6/K48 Ub3 debranching at 30 {degree sign}C is slower than what was reported by others, which varied from 3 to 12‐fold faster at 37 {degree sign}C, whereas the KM values are similar”The authors should note that two entirely different techniques (MS vs probe binding) and temperatures were employed in the kinetics assays, which may underlie differences in observed kinetics. One question for the authors is that given the fact that UCH37 must operate at 37 degrees in humans, where questions of its kinetics competing with proteasomal degradation may reasonably be raised, why did they choose to measure kinetics at 30 degrees? How does this relate to the argument in the discussion that relatively “slow” debranching would not compete with substrate engagement by the proteasome? Would the authors expect this to also be accurate at 37 degrees where UCH37 may be faster as per Strieter et al.?

In Deol et al., steady-state kinetics of K6/K48 Ub3 debranching was measured by SDS-PAGE, not Mass Spec. The rates of debranching of high molecular weight chains, as measured by Mass Spec in Deol et al., were presented as *k*_obs_ and are not directly comparable to the steady-state kinetic measurements; thus, we do not make comparisons to those results.

We routinely perform fluorescence-based assays at 30 °C because (a) fluorescence is temperature sensitive and it is easier to maintain small volumes at a steady temperature at 30 °C than 37 °C; (b) because the reactions were continuously monitored, we needed to minimize evaporation (again, of small volumes) during the time course. We have also performed many gel-based assays that were done at 37 °C (examples shown in Figure 1) and those results were consistent with results obtained with tUI sensor-based assays at 30 °C.

The k_cat_ value we measured is indeed slower than Deol et al., yet the K_M_ values are very similar. We want to point out that this is not uncommon in enzyme characterizations — differences among enzyme preparations or the amounts of active enzyme are more likely to be reflected as Vmax differences. For this reason, we produced four different enzyme preps (Figure 2—figure supplement 1B) and characterized each by Ub-AMC hydrolysis. We tried two different procedures (co-purified NS-UCH37-RPN13C and co-purified NS-UCH37-RPN13FL) to produce identical proteins as described by Deol et al.

Previously, when we and independently Sahtoe et al., had reported the kinetic parameters of Ub-AMC hydrolysis by UCH37-RPN13, we obtained very similar values. Using this criterion of specific activity of Ub-AMC hydrolysis, we are confident that the enzyme activity and quantitation we report are reliable. The k_cat_ values for K6/K48 Ub3 hydrolysis reported by Deol et al. ,varied over a 5-fold range (i.e., 16 min-1 (Figure 3F), 9 min-1 (Figure S3C) and 3 min-1 (Figure S4D)). Without careful analyses using a common substrate (e.g., Ub-AMC), direct comparison of the two studies probably is problematic. In Figure 1D of Deol et al., in which 1 μm UCH37 was used with 10 μm Ub3, the relative intensities of the two Coomassie-stained protein bands are quite close. In comparison, Figure 1A in our paper shows a gel with 1 μm UCH37 and 5 μm Ub3 stained by Coomassie as well; the relative intensities of those protein bands are very different. Currently we are unable to account for these discrepancies.

10) While the biophysical measurements were crystal clear, aspects of the manuscript dealing with proteasomal foci formation and especially RAD23B accumulation were unclear as to what precise conclusion was arrived at by the authors. RAD23B did not appear to show any binding preference in their assays for example and they did not speculate further. These experiments almost felt like an attempt to distinguish the current manuscript from previous literature that had already demonstrated impaired proteasomal function due to UCH37 knockouts. The authors should clarify in the text whether they envision that foci are formed because of the accumulation of polyubiquitylated substrates, or, if foci formation occurs first, which then leads to the accumulation of polyubiquitylated substrates. If it is the former, isn't foci formation simply one of the many physiological outcomes from a blocked proteasome that is still recruiting polyubiquitylated species which in turn recruit more proteasomes? If there is something unique about the clustering of proteasomes in this manner, the authors should definitely highlight this unique aspect of UCH-37 function.

In the absence of an applied stress, proteasome foci are rare, although less rare in the UCH37KO cells (Figure 5D). Upon sucrose or arsenite stress, foci and the consequence of perturbing UCH37 (i.e., UCH37KO, C88A and EWI) become much more obvious. We agree with the reviewer that this most likely reflects impaired proteasome function. However, we cannot differentiate whether foci formation was promoted because of substrates that accumulated prior to the stress or if substrates accumulated after the stress due to poor degradation. In either case, while the importance of UCH37 in stress response might have been predictable, it nevertheless had not been addressed.

For many years, we have known that RAD23B was highly enriched in UCH37-C88A proteasomes. With the report from Deol et al., logically we hypothesized that RAD23B might prefer branched chains. Although it was disappointing to find that RAD23B did not exhibit such a preference with Ub3-size chains, we feel that the result nonetheless will contribute to efforts to better understand the signals and dynamics of stress-induced Ub-rich inclusions. How branched poly Ub promotes substrate degradation by the proteasome, whether branched chains are produced under those stress conditions, and whether they are required for proteasome foci to form all remain open questions. These are especially relevant since the report in 2020 by Yasuda et al., that sucrose-induced proteasome foci depend on RAD23B. Given our finding that RAD23B does not have a branched chain preference, we think it is unlikely that branched chains are required.